# Northern expansion is not compensating for southern declines in North American boreal forests

Ronny Rotbarth [1] ✉, Egbert H. Van Nes [1], Marten Scheffer [1], Jane Uhd Jepsen [2], Ole Petter Laksforsmo Vindstad[3], Chi Xu [4] & Milena Holmgren[1]

Climate change is expected to shift the boreal biome northward through expansion at the northern and contraction at the southern boundary respectively. However, biome-scale evidence of such a shift is rare. Here, we used remotely-sensed tree cover data to quantify temporal changes across the North American boreal biome from 2000 to 2019. We reveal a strong north-south asymmetry in tree cover change, coupled with a range shrinkage of tree cover distributions. We found no evidence for tree cover expansion in the northern biome, while tree cover increased markedly in the core of the biome range. By contrast, tree cover declined along the southern biome boundary, where losses were related largely to wildfires and timber logging. We show that these contrasting trends are structural indicators for a possible onset of a biome contraction which may lead to long-term carbon declines.

Boreal forests are one of the largest terrestrial biomes on Earth, storing one-third of the global terrestrial carbon above and below-ground[1,2] and acting as a critical tipping element of the climate system[3]. Boreal regions have warmed twice as fast as the global average[4]. Consequently, the integrity of the boreal biome is increasingly undermined by climate change[5]. Rising temperatures can directly affect plant growth, mortality, and recruitment[6]. In addition, there is increasing evidence that climate change alters natural disturbance regimes. Droughts, wildfires, and insect outbreaks have increased in extent, frequency, and severity[7–11]. Although these disturbances are an inherent and vital part of boreal forests, intensifying disturbance regimes and human activities could undermine boreal forest resilience, resulting in tree cover loss[6,12–14]. The interactions between these direct and indirect effects of climate change may determine the future health[5] and distribution range of boreal forests[15].

It is generally assumed that the boreal biome, as a whole, will expand northward as climate warming progresses[16] (Fig. 1). At the northern biome boundary, trees may recruit, grow, and survive better under longer and warmer growing seasons[13,17]. As a result, trees are

expected to expand their range into current Arctic shrub and tundra-dominated areas where tree growth has been formerly temperature-limited[18]. Widespread expansion of woody species into Arctic tundra has indeed already been observed[19–25]. Improved growing conditions can also speed up forest recovery following disturbances, such as wildfires or timber logging, and may facilitate forest expansion in the north. In contrast, at the southern boreal distribution boundary, boreal forests may become increasingly stressed as climate change progresses. Warmer and drier conditions may lead to reduced growth in cold-adapted boreal tree species[26,27]. Reductions in growth rates indicate declines in vitality and can considerably increase the risk of mortality[28], a process which may already occur at low rates of warming along the southern boreal boundary[27]. At the same time as boreal forest tree species experience growth reductions, broadleaf tree species associated with temperate forests benefit from warming temperatures[27]. As a result, temperate species may become more dominant in the south, leading to shifts in species compositions[29]. In addition, more frequent wildfires and insect outbreaks under warmer and drier conditions could increase tree mortality in the southern

[1]Environmental Sciences Department, Wageningen University, Wageningen, The Netherlands. [2]Norwegian Institute for Nature Research, Fram Centre, Tromsø, Norway. [3]Department of Arctic and Marine Biology, UiT – The Arctic University of Norway, Tromsø, Norway. [4]School of Life Sciences, Nanjing University, Nanjing, China. ✉e-mail: ronny.rotbarth@wur.nl

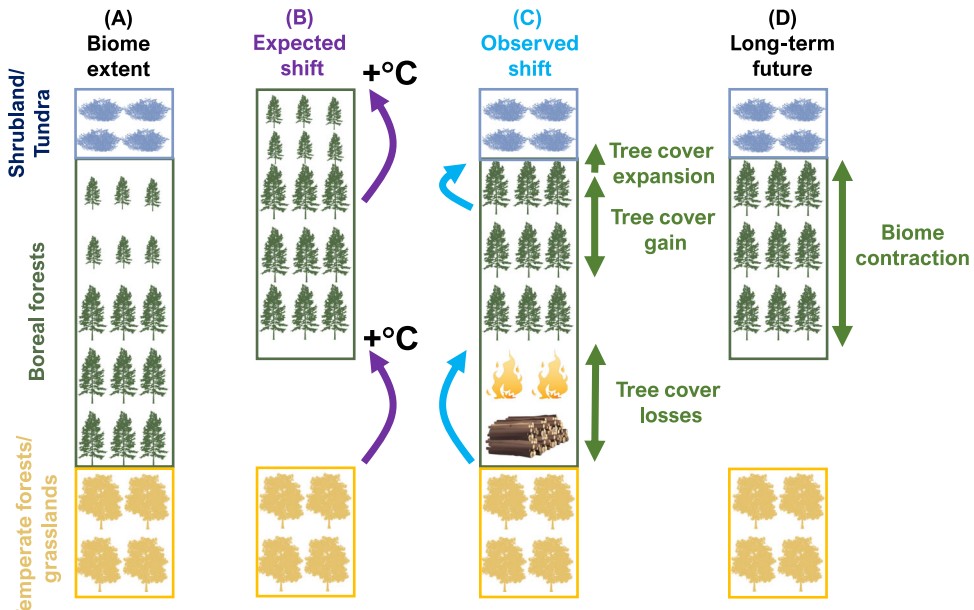

**Fig. 1 | Conceptual overview of hypothesised biome distribution changes.** Each panel represents the latitudinal distribution of the temperate, boreal, and tundra biomes from south (bottom) to north (top). **A** Representation of the current distributions of biomes. **B** The boreal biome is expected to move northward with climate change (purple arrows). In the south, the boreal distribution boundary moves northward through tree mortality and reduced recruitment, owing to deteriorating growing conditions. The northern boundary is expected to migrate through dispersal and colonisation into the warming Arctic shrubland and tundra. **C** Observed shift through changes in tree cover: Tree cover loss in the south is amplified by timber harvests and wildfires. In the north, tree cover within the boreal biome increases by infilling of open areas. However, tree cover expansion into the Arctic is slow. **D** Over time, the fast tree cover losses in the south and the slow expansion in the north lead to a transient contraction of the boreal biome.

boreal. In combination with slower post-disturbance recovery from reduced growth rates, this could further facilitate the expansion of temperate forests[26,27,29] or steppe-grasslands[30,31] into today's southern boreal forests.

Crucially, the processes affecting the northern and southern boreal distribution boundary may also occur at very different rates. The northwards migration of boreal trees into Arctic tundra depends on seed dispersal and successful recruitment of trees in suitable microsites. This process will likely be relatively slow and spatially heterogeneous. It is therefore unlikely that the expansion rate of trees is able to match current rates of climate niche shifts[24,32]. In contrast, the forest contraction along the southern distribution boundary may be relatively fast as increasingly hotter and drier growing conditions and intensifying disturbance regimes accelerate tree mortality events. This asymmetry of slow boreal forest expansion in the north and faster boreal forest contraction in the south would result in a temporary disequilibrium[33] between the distribution of tree cover and its climatically determined niche, thus causing a transient overall contraction of the boreal biome. Whether and to what extent a transient contraction will continue, depends on multiple factors related to the rates of future climate change, northern expansion processes, and southern retreat. Consequently, a boreal biome contraction could follow different trajectories: (1) a failure to compensate by northern biome expansion, especially when non-analogous systems establish in the north or when drivers leading to biome loss in the south accelerate, (2) a compensation by an acceleration of northern biome expansion that matches the pace of southern biome loss and (3) a compensation by a recovery of the biome along the southern margin, if drivers of biome loss decrease in magnitude. Which of these trajectories the boreal biome will ultimately follow and over what timescales trajectories will take place is highly uncertain. Nonetheless, models to predict such biome changes are often based on very long-term dynamics towards a system's equilibrium which may fail to detect transient processes that could be more important over timescales relevant for humanity[34].

A boreal biome contraction could have substantial consequences for the functional characteristics of the biome, for example, the ability of carbon storage and sequestration. The boreal biome stores vast amounts of carbon in the biomass of forests and even more carbon in the often permanently frozen soils[2]. A contraction of the distribution of forests may release parts of this carbon pool, if stress and disturbance-induced carbon losses are not compensated by additional warming-induced carbon gains. A resulting release of carbon would further accelerate climate warming. A change in forest distribution will likely also lead to a change in albedo, i.e., the amount of reflected solar radiation[35]. Forest expansion may reduce albedo which would lead to an increase in local and regional surface warming. Contrarily, a forest reduction or a shift from darker coniferous to lighter-coloured broadleaf tree species can increase albedo and have a cooling effect. As warming and cooling effects of albedo can interact with effects from changes in the carbon balance[36], it is crucial to understand boreal forest dynamics with regards to a shift and possible contraction of the biome.

A biome contraction through the loss of associated ecosystems may play out over very long time periods. The direct detection of such slow processes may therefore be difficult. Moreover, the observation of an ongoing contraction may limit possible mitigation measures. However, dynamics in forest structural properties can be quantified on a biome scale. Depending on the spatial distribution changes of such dynamics over time, they can serve as indicators for whether and to what extent tendencies towards a biome contraction exist.

Multiple studies have investigated boreal forest dynamics through the use of vegetation indices as a measure of productivity[21,37–39]. Most of these studies conclude that spatial variation in productivity trends was high but that productivity increased across large parts of the boreal biome, mainly in the north, and decreased at the warmer southern biome margin[37,39]. However, to observe a possible shift and transient contraction of the boreal forest distribution, interpretable rates of change are necessary. In this respect, changes in

vegetation indices are often challenging to interpret because signals of productivity change from ground vegetation and canopy forests cannot easily be distinguished, even though spectral greenness indices have been related to field-based observations[37]. In addition, combined spectral greenness data from multiple sensors may create false signals from calibration[38,40]. Hence, the physical characteristics of forests may be better measures of forest distribution changes.

We, therefore, use satellite-based, annual time series of tree cover from the Moderate Resolution Imaging Spectroradiometer (MODIS)[41] to quantify changes in the distribution of the boreal biome across North America (Canada and Alaska) over the past two decades (2000–2019). Tree cover is linked to various functional characteristics, such as biomass, carbon storage, albedo, and forest structure[42]. As a physical metric of forest extent, changes in tree cover are also more easily interpretable in the context of biome distribution changes than spectral greenness indices. We evaluate tree cover changes with respect to their relative distance to the northern and southern boreal biome boundaries[5] and assess how these dynamics at the edges of the distribution range depend on the interactions between climatic conditions[43] and local disturbances (wildfire[44] and timber harvest[44,45]) for different land cover types[46].

We aim at testing the following three specific hypotheses: (1) Contrasting tree cover changes occur at the northern and southern boundaries of the boreal biome, consistent with a northward boreal biome shift. We expect these differences to be characterised by relatively slow tree cover gains at the boreal-tundra boundary and relatively fast losses at the southern boundary. This mismatch would also be coupled to a spatial range shrinkage of tree cover distributions in the south. (2) The extent of tree cover change depends on local disturbance regimes (wildfires and timber harvest) and land cover types. We expect disturbances to accelerate tree cover losses and that these losses are stronger in the southern than in the northern boreal regions. We expect needleleaf forests in the south to sustain greater tree cover losses than broadleaf or mixed forests, as the latter has been shown to perform better under warming conditions than needleleaf species[27]. (3) Tree cover changes are related to temperature and precipitation and their trends across the past decades. We expect that tree cover gains tend to occur more in cold, moist regions and tree cover losses in warm, dry regions. Changes towards warmer and drier conditions will accentuate tree cover losses. We demonstrate an asymmetry in tree cover change between the northern and southern boreal forests, leading to a range shrinkage of tree cover distributions. We interpret these dynamics as an onset of a potential biome contraction.

## Results

### Tree cover changes across the boreal forest biome
We calculated both absolute tree cover changes and changes relative to the mean tree cover of all years between 2000 and 2019 (Tree cover$_{rel}$ = $\frac{\text{Tree cover change}}{\text{Mean tree cover}(2000-2019)}$, see Methods for details). Both approaches revealed an asymmetry in tree cover changes at the northern and southern boreal forests in North America (Fig. 2). Our analysis of ~13,000 sample plots along south-north transects across the North American boreal region (see Methods) showed that across the North American boreal biome tree cover increased, albeit with a high variation (mean change of 0.12% ± 0.40% standard deviation SD per year). We found considerable regional variability in how tree cover has changed over the past two decades. Tree cover loss was the dominant trend around the southern boreal boundary (mean tree cover loss of 0.13% ± 0.39% SD per year). When moving northwards, we observed a clear tendency towards tree cover gains of gradually increasing magnitude. These tree cover gains were strongest in the northern half of the boreal interior with a mean increase of 0.22% ± 0.4% SD per year. At and beyond the current northern boundary, tree cover changes were low, with a 0% ± 0.07% SD per year change north of the boundary. The biome-wide tree cover increase was

mainly driven by increases in the northern interior of the current distribution range. The observed regional changes in tree cover resulted in a range shrinkage of tree cover distributions and a reduction in treed area around the boreal biome boundaries (Supplementary Figs. 1–5 and Supplementary discussion). Thus, North American boreal forests declined in latitudinal distribution range driven by southern losses but became denser due to the tree cover gains in the northern interior.

The north-south asymmetry was also observable in tree cover changes in different ecozones. We used the latest map of ecozones from the Canadian Ecological Framework[47] and calculated mean trends for each ecozone based on our sample plots (Supplementary Fig. 6). Ecozones located in the southern boreal biome or at the boreal-temperate transition zone either lost tree cover or had net-zero change, e.g., −0.1% ± 0.39% SD per year in the Montane Cordillera, 0% ± 0.5% SD per year across the Boreal Shield or +0.02% ± 0.43% SD per year on the Boreal Plains. Contrarily, northern and boreal interior ecozones or those of higher elevations had the highest tree cover increases, e.g., Taiga Plains with 0.27% ± 0.39% SD per year or the Boreal Cordillera with 0.26% ± 0.37% SD per year. Ecozones beyond the boreal boundary did not change much, e.g., the Southern Arctic had a mean tree cover change of 0% ± 0.09% SD per year. Across Alaskan boreal forests, tree cover increased slightly by 0.08% ± 0.37% SD per year.

Because the patterns of absolute and relative tree cover change are similar (Supplementary Fig. 7), we hereafter focus the presentation and discussion of results on absolute tree cover changes.

### Tree cover dynamics of disturbance regimes and vegetation types
North American boreal forests are subject to strong disturbances, both natural and anthropogenic[5,48]. On a continental scale, the dominating sources of disturbance are wildfires and timber harvest[49]. We therefore compared tree cover changes in areas affected by wildfire[44,50,51] and areas that were harvested for timber since 1985 (timber harvest only for Canada)[44]. We hereby considered the year when disturbances occurred and distinguished between disturbances occurring within (2000–2019) and prior to (1985–1999) our study period. These were contrasted to areas with no documented disturbance after 1985.

Disturbances had a marked impact on tree cover change across the boreal biome (Fig. 3) and were associated with most observable tree cover losses. In the absence of fire and harvest since 1985, boreal tree cover increased across most of the boreal distribution range with a mean change of +0.16% ± 0.24% SD per year. This increase was lower in the south than in the north (+0.04% ± 0.28% SD per year at the temperate-boreal transition zone and +0.24% ± 0.25% SD per year in the northern interior, Fig. 3b and Supplementary Fig. 6 for ecozones). Where wildfire and timber harvest occurred since 2000, tree cover decreased across most of the study area (Fig. 3). These losses were higher in areas affected by timber harvest than by wildfire, with mean changes of −0.15% ± 0.43% SD and −0.05% ± 0.48% SD per year, respectively. Losses of tree cover in disturbed areas were highest in the southern boreal where wildfires and harvests overlap in extent (Fig. 3 and Supplementary Fig. 5). In the temperate-boreal transition zone, harvests dominated tree cover losses (−0.3% ± 0.42% SD per year compared to −0.12% ± 0.33% SD per year from fires), while wildfires caused higher losses in the southern interior (−0.13% ± 0.5% SD per year compared to −0.06% ± 0.36% SD per year, also compare Supplementary Fig. 6 for ecozones). Interestingly, tree cover in northern boreal forests increased in our sample plots despite the occurrence and greater extent of wildfires in these parts compared to the southern boreal (Fig. 3, Supplementary Fig. 5; also Supplementary Fig. 6 for ecozones). Areas disturbed between 1985 and 1999 were associated with the highest rates of tree cover gains over the past two decades. In particular, burnt areas showed the highest tree cover increase across all disturbance categories with a mean 0.4% ± 0.35% SD increase per

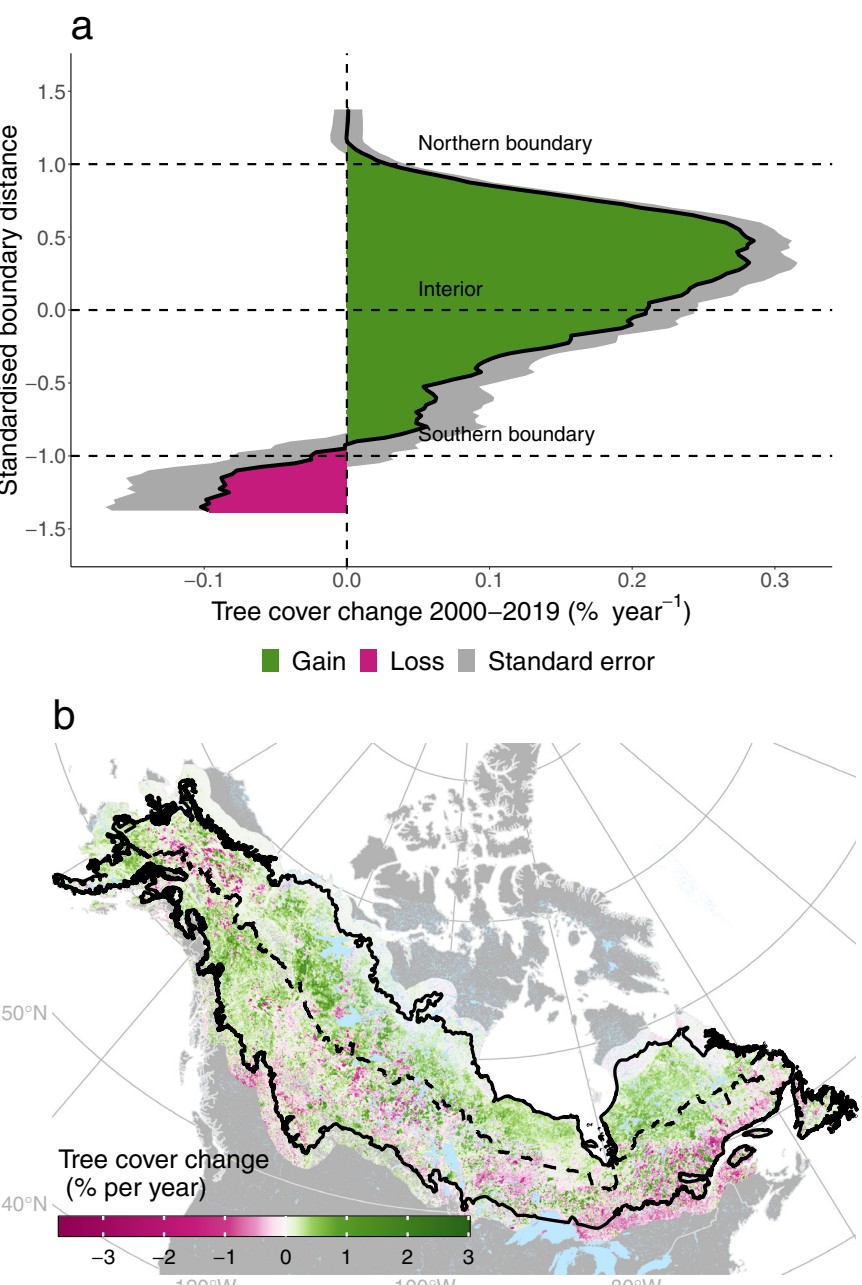

**Fig. 2 | Absolute tree cover changes of North American boreal forests from 2000–2019. a** Tree cover changes based on the MODIS VCF tree cover product of 13,000 sample plots along 69 south-north transects (see Methods). Tree cover change was calculated using Theil-Sen's slope estimations. The horizontal lines indicate the northern and southern boreal boundary based on Gauthier et al. 2015[5]. Standardised boundary positions were calculated for each sample plot based on its relative position within the boreal boundaries. For better visualisation, tree cover changes are shown as running means along boundary distances with bins of 0.025 year. Tree cover gains in areas logged between 1985 and 1999 were

(original results in Supplementary Fig. 7). **b** Mapped tree cover changes across the entire North American boreal forest biome, extending 120 km around the boreal biome boundary (black solid line, based on Gauthier et al. 2015[5]). The dashed line represents the halfway positions between the northern and southern boreal biome boundary (comparable to a standardised boundary distance of 0 in **a**). For a better visibility, the map is of coarser resolution than the MODIS data set (-1000 m instead of 250 m, rescaling was done through cell aggregation using means).

year. Tree cover gains in areas logged between 1985 and 1999 were lower than in areas that burned during the same period (+0.16% ± 0.41% SD per year). At the temperate-boreal transition zone, tree cover losses occurred under timber logging (−0.04% ± 0.34% SD per year) and, to a minor extent, under wildfire (−0.01% ± 0.24% SD per year, Fig. 3b).

The patterns of tree cover change along the boreal distribution gradient may differ across vegetation types, as needleleaf, broadleaf, and mixed forests may respond differently to climatic conditions and

climatic changes. In this context, we compared tree cover changes across different vegetation types[52], dominated by non-woody vegetation (herbaceous plants and mosses), shrubs, needleleaf, broadleaf, and mixed forests. We found the asymmetry in tree cover changes along south-north gradients to be persistent across many vegetation types and disturbance types (Fig. 4). Hereby, vegetation type determined at which location within the boreal interior tree cover gains peaked. Needleleaf forests, for example, gained most tree cover in the northern boreal, while trees growing within areas of non-woody

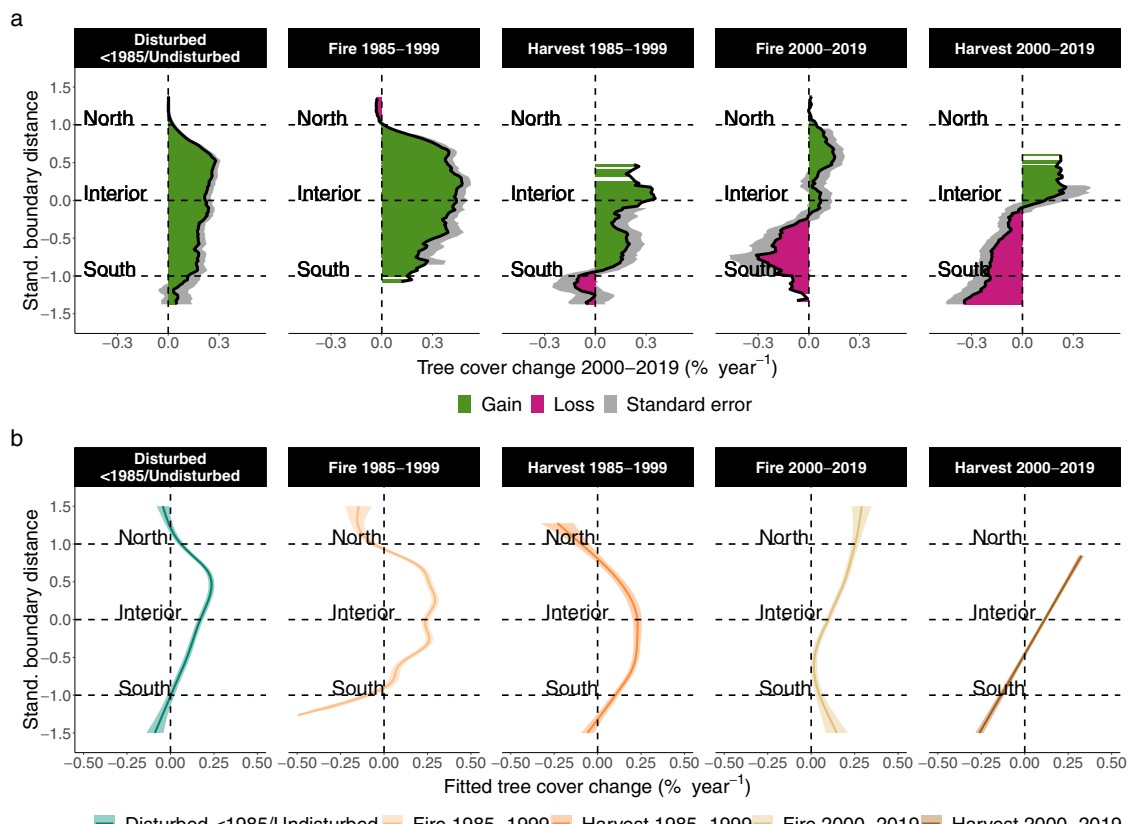

**Fig. 3 | Tree cover change of North American boreal forests between 2000–2019 by disturbance type. a** Tree cover changes by disturbance type expressed as running means along south-north boundary distances with bins of 0.025 (original results in Supplementary Fig. 16a). Boundaries were derived from Gauthier et al. [5]. **b** Fitted tree cover changes from generalised additive mixed-effects models. Data are presented as model fits ± standard errors around the fitted lines. Standardised boundary distance was included as fixed smoothed explanatory variable in the model and fitted using disturbance type as interaction term. Tree cover changes were calculated as described in Fig. 2. Disturbances by wildfire and timber harvest cover the periods 1985–1999 and 2000–2019 and were derived from the CANLaD data set (see Methods for details). Undisturbed forests are defined as areas unaffected by fire and harvest since 1985. Relative tree cover changes by disturbance type are shown in Supplementary Fig. 16b. Fitted model lines including raw data points are shown in Supplementary Fig. 7.

vegetation or shrubs had the strongest increases in the interior. In mixed or broadleaf forests, tree cover changed approximately linearly along boreal boundary distances, characterised by tree cover gains towards the north and losses with closer proximity to the south. This is with the exception of mixed forests which were logged prior to 2000 and which peaked in tree cover gains in the southern boreal.

## Tree cover dynamics along climatic and other environmental gradients

We further analysed how tree cover changes were related to mean annual temperatures, mean annual precipitation, and their trends across two time periods (1980–2019 and 2000–2019). We also considered elevation and mean tree cover (2000–2019) in these analyses as climate patterns and trends change with elevation and changes in tree cover may be related to initial tree cover. We performed model fitting separately for each disturbance type (i.e., undisturbed areas and areas affected by wildfires and logging), as disturbances may alter the underlying relationship between tree cover change and climate. For example, some areas may have favourable climatic conditions for tree cover increases but experience tree cover losses due to fire occurrence or timber harvests. As we used a Canadian dataset of disturbances, we restricted our analyses to the Canadian boreal forests.

The humpback pattern in tree cover change along standardised boundary distances (Fig. 3b) was also evident along climatic and tree cover gradients: Considerable tree cover gains occurred in areas with moderate mean tree cover (~20–40%, Fig. 5a), intermediate

temperature ranges (-7–0 °C, Fig. 5b) and low to moderate amounts of precipitation (~400–1000 mm, Fig. 5c), all of which correspond to the interior and northern interior of the boreal biome distribution (Supplementary Fig. 8). In contrast, open and dense forests, very cold and very warm areas, as well as dry and wet areas, either lost tree cover or experienced little change. These patterns of tree cover change were persistent across disturbance regimes and disturbance timing. Tree cover also increased considerably in areas that are undisturbed since 1985 and which also experienced moderate long-term and short-term warming (~0.04 °C annual warming since 1980 or ~0.1 °C annual warming since 2000, Supplementary Fig. 9e, g). Neither recent nor long-term trends in precipitation showed clear relationships with tree cover change (Supplementary Fig. 9f, h). Elevation had little effect on absolute tree cover changes (Supplementary Fig. 9b, but compare with Supplementary Fig. 10c).

## Discussion

Our continental-scale analysis revealed that North American boreal forests have experienced a marked north-south asymmetry in tree cover changes over the past two decades, characterised by slow gains at the tundra-forest boundary and fast losses at the boreal-temperate boundary (Fig. 2). This suggests that the northern expansion rate of the boreal biome has, so far, been unable to keep up with the rates of forest decline in the southern edge of the biome distribution. This indicates that climate change has induced a state of disequilibrium[33] between the potential and realised distribution of tree cover in the

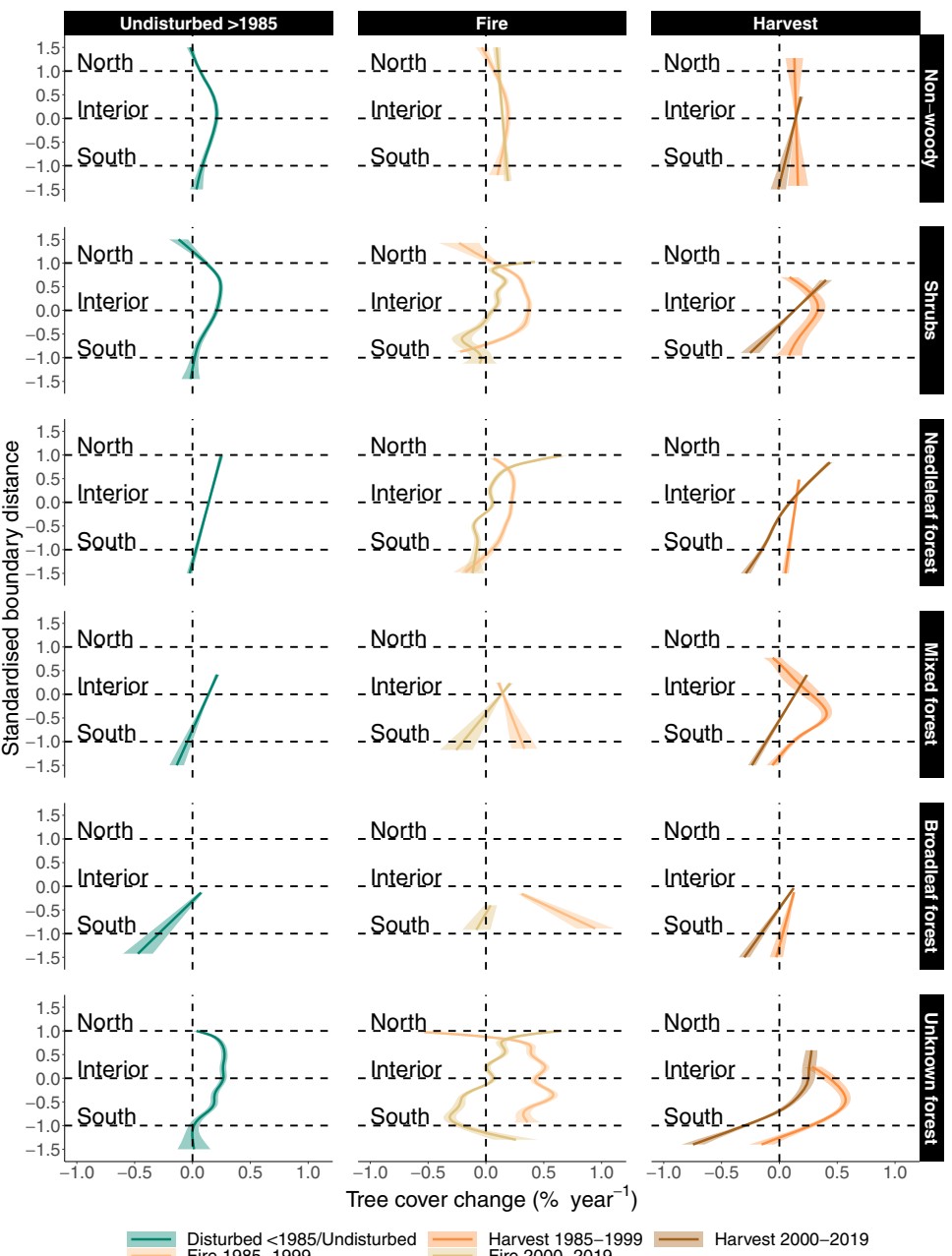

**Fig. 4 | Relations between tree cover change, disturbances, and vegetation type between the southern and northern boreal forest biome boundaries.** Tree cover changes are visualised along standardised boundary distances from south (−1) to north (1) as model fits ± standard errors around the fitted lines from a generalised additive mixed-effects model (see Supplementary Fig. 17b for relative tree cover change). We included standardised boundary distance as fixed smooth explanatory variable in the model. We further included interaction effects within the smoothing term considering 30 different combinations of disturbances (panel columns and colours) and vegetation types (panel rows). We accounted for spatial correlation in the model and treated sample transects as random effects. Horizontal dashed lines represent the southern, interior, and northern boreal forest boundaries. Fitted model lines including raw data points are shown in Supplementary Fig. 17a.

North American boreal biome, opening up the potential for a transient biome contraction. While our study period is too short to detect a direct loss of the boreal biome associated with a contraction, the mismatch between a lack of tree cover expansion in the north and the decline in the south are structural indicators for a possible onset of such a contraction. This is supported by additional lines of evidence showing a clear range shrinkage of tree cover distributions along the biome boundaries (see Supplementary Figs. 1–5 and Supplementary discussion). We also highlighted that this shrinkage may occur despite our observation that forests became denser, especially in the northern interior (Supplementary Fig. 1). This means that the overall biome may

have increased in surface area through infilling of available space in the northern boreal but declined their latitudinal distribution range.

The absence of tree cover expansion at the boreal-tundra boundary could stem from a lag in tree responses to climate change[33,53] which may be related to dispersal limitations, the influence of disturbances (such as Arctic fires or water logging from permafrost collapse), poor nutrient availability[54,55], slow adaption or herbivory[56]. Lags are likely to be present at the boreal-temperate boundary as well, despite the higher rates of change (i.e., losses) compared to the northern boundary. A delay in mortality of long-lived species such as trees, genetic adaptations to changes in climate, or anthropogenic

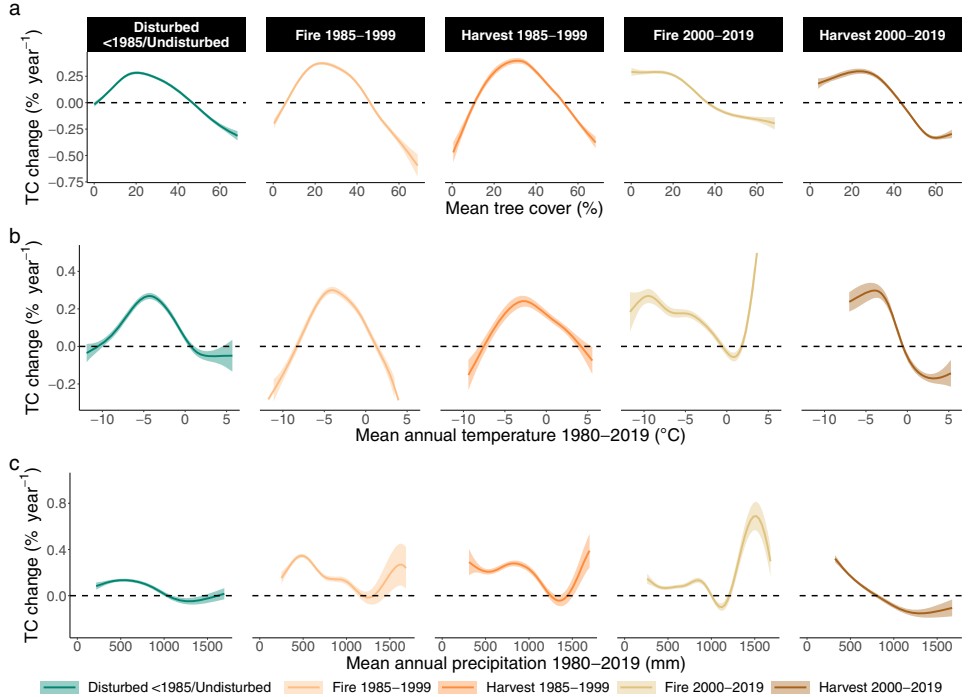

**Fig. 5 | Relationships between tree cover (TC) change 2000-2019 across North American boreal forests and environmental conditions. a** Mean tree cover 2000–2019. **b** Mean annual temperatures (MAT). **c** Mean annual precipitation (MAP). Relationships are shown as model fits ± standard errors around the fitted lines for different disturbance categories (colours). Model fits were generated from generalised additive mixed-effects models. Each predictor of tree cover change was included in a separate model due to correlations between predictors. Transects were used as random effects in the models. Additionally, tree cover change was fitted by disturbance type and assuming an exponential spatial correlation structure. Other environmental conditions are shown in Supplementary Fig. 9. Fitted model lines including raw data points are shown in Supplementary Fig. 18. Absolute tree cover change was calculated through Theil-Sen's slope estimation method and is expressed in absolute terms (see Supplementary Fig. 19 for relative TC change). Climatic variables were taken from the ERA5 Monthly Averaged data set and cover the period 1980–2019 and 2000–2019.

influences (e.g., through tree planting) can obscure even higher rates of tree cover losses than we report here[53].

Our results add to research showing a boreal biome shift from changes in spectral vegetation greenness[37]. Results on trends of greenness indices suggest a northward shift of the boreal biome, with browning trends more dominant in the south and greening trends in the north. While these results are consistent with ours, by using tree cover as a physical measure of forest characteristics, changes in tree cover we observed are interpretable in the context of a biome shift. By adding the magnitude of change, we were also able to demonstrate the marked differences in rates of change between the northern and southern boreal biome. We interpret these differences as a range shrinkage of tree cover distributions and an indicator for a possible future biome contraction. Future trajectories of these transients will strongly depend on the speed of climate change and its effects on tree growth and disturbance regimes. Long transients[34] may thus also be possible, meaning that compensation of southern forest loss by northern forest expansion would not happen over human timescales.

Wildfires and timber harvests were associated with the magnitude and direction of tree cover changes across most of the boreal biome distribution (Fig. 3). Especially in the southern parts of the boreal biome, tree cover losses were highest in regions affected by disturbances over the past two decades. Our results are in line with other research identifying the considerable contribution of disturbances to forest loss[21,57,58]. Our observations also indicate that forests in the southern boreal biome did not recover from wildfire and timber harvests within the 20-year study period[57–59]. Although southern boreal forests were exposed to fewer and smaller wildfires than northern forests (Supplementary Fig. 11), tree cover losses were larger here than at the boreal interior. These observations suggest slower recovery of

southern boreal forests following wildfire perturbations in the south[59–61]. This is further supported by the lower tree cover gains in areas of the southern boreal which burned prior to our study period between 1985 and 1999 and those with an unknown disturbance prior to 1985 (Fig. 3b). We, therefore, suggest that observed declines in growth rates of boreal tree species in the southern biome[26,27] lead to a considerably reduced post-disturbance recovery ability. In contrast, tree cover increased in the northern interior range of the boreal distribution despite more frequent and larger wildfires in the north than those occurring in the southern boreal range (compare Fig. 3 and Supplementary Fig. 10). These tree cover gains in the presence of recent wildfires seem counter-intuitive. However, as burnt areas rarely covered our sample plots completely but were rather localised patches, tree cover gains in undisturbed areas surrounding those patches could have compensated the losses incurred by fire. In addition, faster post-fire recovery in the northern interior may have accelerated such compensation. Our results suggest that such compensating processes may have occurred in the boreal interior but not in the southern boreal where tree cover losses prevailed. Similarly, tree cover gains in areas that have burned between 1985–1999 in northern regions were higher than in the south, indicating a much faster recovery[57–59]. This faster post-fire recovery in northern regions may result from several processes. Here, climate change may have made growing conditions more benign for post-disturbance tree growth than in the southern distribution range. Alternatively, faster recovery may be driven by post-fire changes in species composition, from slow-growing coniferous to fast-growing deciduous species[62]. These compositional changes may partly be reflected in the tree cover gains we observed following fires in the northern boreal interior.

Tree cover changes were also related to timber harvests (Fig. 3). Logging operations are concentrated at the southern boreal interior and the boreal-temperate boundary[63,64] (Supplementary Fig. 11), contributing to the tree cover losses we report over the past two decades Similar to wildfire, tree cover in logged areas increased when moving northward which could be a result of the lower harvest extent in these parts. Changes in tree cover for areas harvested between 1985 and 1999 showed similar patterns as for wildfire with lower tree cover gains in the south than in the north. However, the successional trajectories which determine the recovery from disturbance likely differ strongly between wildfire and timber harvest. Forest management of areas under timber logging in Canada have to ensure forest regeneration through either natural regeneration or active planting of tree seedlings. Hereby, replanting pre-determines the recovery of forests after logging and therefore may have affected the dynamics in tree cover change we observed. Despite the similar patterns we found for tree cover changes in the southern and northern boreal in burnt and logged areas, the underlying mechanisms governing these dynamics are likely different. Regarding pre-harvest recovery, our results mean that assisted regeneration of forests through planting may nevertheless be equally affected by changing growing conditions as areas recover from wildfire. To what extent regeneration success from replanting after logging differs between southern and northern logging areas is beyond the scope of our study.

Our results highlight that disturbances are linked to the observed range shrinkage of tree cover distributions. Boreal forests at their southern boundary may be limited in their ability to recover from disturbances compared to their northern distribution. In particular, where wildfire occurrence and timber harvests overlap (Supplementary Fig. 11), the interaction of these disturbance types could amplify tree cover loss[11] and erode forest resilience to future climate change and to intensified disturbances[65]. This is particularly concerning, as wildfires are expected to increase in magnitude, intensity, and frequency in boreal landscapes with further climate change[59–61]. They could thus not only lead to greater tree cover losses but also further offset or stall recovery processes[66,67], eventually amplifying the biome contraction.

In addition to disturbances, we found associations between tree cover changes and climate, indicating that climate and climate change may be a driver of tree cover dynamics over the past decades. Tree cover gains were highest in areas of intermediate temperature and tree cover, typically associated with the boreal interior. These gains could represent an infilling of available spaces by already established trees. Trees in these higher latitudes and higher altitudes may benefit from lifting growth limitations[18] and from shifting climatic niches towards the optimal range for boreal tree species[26]. Our observations are in line with results from previous studies showing improved growing conditions and productivity of northern boreal forests as a result of warming[21,26,37,38,68], albeit with large regional variability in growth responses which can occur due to differences in water availability[69].

In contrast to the boreal interior, tree cover gains at the cold boreal-tundra boundary were slow. The non-linear relationship between tree cover change and temperatures may suggest processes leading to a disequilibrium between boreal biome distribution and shifting climatic niches. Studies on northward forest expansions have shown that forests do not consistently follow their climatic niche everywhere along the northern boundary[19,24,70,71]. Tree recruitment and growth are very slow processes at high-latitude ecosystems. Forest expansion can be locally slowed down by seed dispersal limitation. However, field experiments show that once tree seedlings germinate, their survival strongly depends on the facilitative effects of shrubs ameliorating the harsh abiotic conditions of the far north[72]. Although there is limited evidence that climate warming has facilitated northward shrub expansion[73] and this could further pave the way for trees to follow, climate warming has also increased permafrost collapse. Field

and remote sensing observations show that surface water accumulation from permafrost collapse leads to the browning of shrubs and forests[74,75] and may thus limit forest expansion[75].

In the southern boreal, we observed a tendency towards tree cover losses in areas of warm temperatures even in the absence of wildfires and logging over the past four decades. Our results indicate that deteriorated growing conditions may have reduced tree cover at the warm biome margin, consistent with previous research using greenness indices[37]. Exceeding temperature thresholds under continued warming can lead to growth reductions of various tree species[26,27]. As these thresholds are likely to be reached earlier at the warm edge of the southern boreal biome than in the north, warming could lead to elevated mortality[27,76] and explain the tree cover losses we report here.

Some of the losses at the southern margin could be compensated by an expansion of temperate forests into the boreal biome[15]. We expected that differences in tree cover change between broadleaf and needleleaf forest types would be indicative of a future climate-induced shift in dominance of these groups[29,77–79]. However, we did not find sufficient evidence to support this expectation. Mixed and broadleaf forests generally lost tree cover at the southern boundary similar to needleleaf forests (Fig. 4). While trees growing in areas dominated by shrubs, non-woody vegetation, and needleleaf forests maximised tree cover gains at different locations within the biome, we did not observe such differences for mixed and broadleaf forests. Whether and where a temperate forest expansion into the current boreal biome will occur is still unclear but is unlikely to occur rapidly[53].

The observed tree cover changes have implications for carbon storage, regional and global climates, biodiversity, and people's livelihoods and could initiate or alter feedbacks to other tipping elements of the Earth system[3]. If changes in tree cover can broadly be associated with above-ground carbon storage of boreal forests, our results suggest carbon uptake in the interior of North American boreal forests and a release of biomass carbon in the south over the past 20 years. The infilling of available space by tree cover in the northern interior outweighed the losses in the south and led to an overall increase in treed area and thus biomass carbon. This is consistent with observed overall biomass accrual across Canadian boreal forests over the past three decades[58].

We stress, however, that the short-term increases in biomass carbon based on our results may not be indicative of the long-term net carbon balance of boreal forests, especially under a biome contraction scenario. A reduced recovery ability of boreal forests from disturbance at the southern margin could prolong the time until carbon losses are compensated by carbon gains[80] and may lead to non-linear transitions and sudden loss of tree cover and carbon[65]. Further increases in anthropogenic pressure from future climate-induced expansions of population densities[81] and agricultural areas[82] into the southern boreal zone may accentuate a boreal contraction. Additionally, the loss of southern boreal carbon due to climate change and disturbances and its potential replacement by low-carbon systems (e.g., shrublands or grasslands) could outweigh carbon gains in the north[83]. The latter may happen at the expense of thawing permafrost and an associated release of carbon from the extensive permafrost carbon pools[2,84]. Finally, the extent to which biomass carbon can increase in the northern boreal biome is limited by the available space and will eventually saturate over time, while a retreat of the southern biome boundary may continue. Whether biome carbon gains can outpace carbon losses is outside the scope of our study. The uncertainty surrounding this question, however, highlights the importance of long-term monitoring coupled with accurate predictions from climate or dynamic vegetation models.

Tree cover changes additionally impact albedo and can alter regional warming and interact with permafrost processes[17]. The tree cover gains we report here for the northern interior are likely to

decrease albedo which may lead to additional warming[85] and a reduction of the climate effect through carbon uptake[36]. Contrarily, losses of tree cover in the southern boreal biome or a replacement of conifers by broadleaved trees can increase albedo and may have a cooling effect[86]. Whether and how a contraction of the boreal biome causes warming and cooling and to what extent these effects can influence global climate is subject to future research.

While satellite-derived tree cover data have enabled a biome-wide study on a continental scale over the past two decades, tree cover change, especially close to the forest-tundra boundary, may not be captured by the MODIS tree cover data due to the inherent limitations in terms of the applied tree height threshold[87]. A validation of remote sensing data with field-based observations (e.g., from forest inventories) is needed to correct for these limitations. In addition, new remote sensing data (e.g., radar, lidar, and hyperspectral sensors) with higher spatial resolutions are promising tools for understanding biome-wide changes in tree cover and other forest characteristics in high-latitude ecosystems[88]. While their temporal or spatial coverage is still insufficient to replicate our analysis, they may provide valuable insights in the interpretation of the results we presented here, especially with respect to cross-sensor validation.

While wildfire and timber harvest are the most important drivers of boreal forest change in North America[49], regional disturbances, such as insect outbreaks, have been shown to cause considerable carbon losses[89]. These disturbances may become more prevalent in the future[90] and could have amplifying effects, when they occur in combination with other disturbance types[11,91]. Accurate estimations of forest damage by these additional disturbances on biome scales are, however, still rare.

Lastly, we argued that our observations of tree cover change may be evident of transient dynamics of the boreal biome. However, data on tree cover dynamics spanning more than the relatively short period of 20 years of the MODIS dataset would highly increase the confidence in the transient nature of observation. Hence, longer-term monitoring of remotely-sensed changes in tree cover and other biome structural characteristics are crucial to identify transient dynamics over decadal and centennial timescales which may differ greatly form their equilibrium state[33]. This is particularly relevant, when long transients prevail in biomes that are subject to slow processes or repeated disturbances[34]. The identification of transient dynamics can correct model predictions of future climate change and vegetation shifts.

In summary, we have presented evidence for an asymmetry in tree cover change between the southern and northern distribution boundaries of North American boreal forests over the past two decades. This mismatch in rates of tree cover change could indicate a tendency towards a possible transient biome contraction. In addition, disturbances and temperatures were strongly related to the observed range shrinkage in tree cover distributions. As disturbance pressure and warming in the boreal biome are unlikely to decline in magnitude in the future, the asymmetry in tree cover change may ultimately lead to a replacement of the southern boreal biome by other systems, e.g., temperate forests or grasslands. The variability in tree cover dynamics highlights that a potential future biome contraction will likely vary considerably in space. It is crucial to understand these future biome dynamics, as a boreal contraction would influence a diversity of ecological processes that affect carbon storage. Our results contribute to the idea that climate change may lead to shifts of entire biomes and that such shifts may not match their climatic equilibrium, leading to transient dynamics. The rate and duration of these shifts are hereby a crucial measure to better predict vegetation and climate change. Long-term studies, combining remote sensing and field observations, are needed to evaluate the trajectories of the entire boreal biome at the northern and southern boundaries and the implications for carbon balance, climate feedbacks and biodiversity.

## Methods

### Study area and sampling design

We conducted our research on a continental scale in North American boreal forests. This region of the global biome distribution has been well-studied and data on climate, disturbances, vegetation, and biomass storage are available. While the boreal biome can be separated by distinct ecoregions based on a broad range of climatic and vegetation conditions, tree cover is located along a distinct south-north temperature gradient, with moderate to high tree cover at higher mean annual temperatures in the south, and a decline in tree cover with declining temperatures poleward[48]. We used this gradient to inform our sampling design.

We collected data within sample plots of 0.05° × 0.05° in size. The plots were located along 69 randomly selected south-north transects (Supplementary Fig. 12). Transect locations were, however, restricted by a minimum distance between them to account for spatial autocorrelation between transects (see Data Analysis for more details). Transects were drawn in such a way that they crossed the northernmost and southernmost boreal biome boundaries and extended beyond them by 20 sample plots (i.e. -120 km). We defined the boreal biome boundary following the delineation by Gauthier et al.[5]. As this definition is based on a combination of climatic conditions, topography, vegetation classes, and forest extent, areas that may not strictly be defined as forest fall within these boundaries, e.g., sparse tundra woodlands. These areas, together with forests, are of interest to our research aims. Consequently, we deemed this data set more appropriate than tree line estimations or forest masks.

Within each plot, we calculated means of all variables which enabled a meaningful comparison of all data based on a single-size unit. To avoid inclusion of tree cover signals from non-natural or mainly unvegetated sources, we excluded pixels classified as urban, cropland, wetlands, bare ground, water, and snow/ice (see Methods for information on classifications). We also excluded all plots in which the aforementioned vegetation types covered the largest area, leaving plots of six main vegetation types: Non-woody vegetation (including herbaceous vegetation, mosses, and lichen), shrubs, needleleaf forests, mixed forests, broadleaf forests, and unknown forests. We extracted data from the remaining 12,954 plots, covering an area of 26.3 million ha (-7% of North American boreal forest extent[48]). We ensured that our sampling design is representative of the disturbance extent and spatial coverage (Supplementary Fig. 13 and Supplementary Table 1).

### Tree cover estimates and change

We used remotely sensed tree cover data from the MODIS Vegetation Continuous Field (VCF) version 6 product for the years 2000–2019 with a spatial resolution of 250 m[41,87] (Supplementary Table 2). We calculated tree cover changes over this period and for each sample plot through Theil-Sen's slope estimation using the zyp package in R (version 0.10-1.1)[92]. To account for potential temporal autocorrelation in annual tree cover estimations, we followed Yue-Pilon's pre-whitening method within this package[93]. The output of this analysis is an annual rate of absolute tree cover change expressed in % year$^{-1}$. We found that absolute change depended on mean tree cover across our study period. To account for this dependency, we additionally calculated relative tree cover change by dividing the absolute value by mean tree cover of all years between 2000 and 2019, which resulted in a change expressed as %$^{-1}$ year$^{-1}$. Absolute and relative tree cover change have separate implications for boreal biome change. Absolute values can be an indicator of biomass change (although both may not always be correlated), while relative values indicate a change with respect to initial conditions. We, therefore, included both measures in our study and performed analyses separately for absolute and relative change. As our main aim was to gain a complete picture of tree cover

changes across the North American boreal biome, we included trend values irrespective of their statistical significance.

## Environmental data
We related observed tree cover changes to disturbance types, vegetation classes, climatic, and other environmental data using several sources summarised in Supplementary Table 2 and shown in Supplementary Fig. 14.

**Disturbance data.** We extracted disturbance data from two datasets covering the period 1985–2019: (1) CanLaD dataset of Canadian wildfire and timber harvest[44] and (2) MTBS dataset of Alaskan wildfires[50,51]. Both datasets include the occurrence and year of disturbance with a spatial resolution of 30 m. To relate disturbance types to our sample plots, we calculated percentage disturbed area within each plot for wildfire and harvest (only for the Canadian dataset). We then classified plots as either Fire or Harvest based on largest disturbed area and recorded the year of that disturbance. Hereby, we classified disturbed sample plots whenever there was any sign of either wildfire or timber harvest, irrespective of its spatial extent within the plots. We then distinguished between disturbances of the following five categories: (1) wildfire within our study period (2000–2019), (2) timber harvests 2000–2019, (3) wildfire 1985–1999, (4) timber harvest 1985–1999 and (5) areas that were either undisturbed or disturbed prior 1985.

**Vegetation classification.** We used the Copernicus Global Land Cover version 2.0.2 map (GLC) with a spatial resolution of 100 m for the year 2015 for vegetation classification[46]. This map can be considered a static land cover map, in which temporary changes in land cover (for example, from forest to herbaceous vegetation after fire) are excluded, unless they remain permanent. Static maps have commonly been used to attribute boreal forest characteristics or change to land cover types and vegetation functional groups[21,37,65], as permanent land cover changes (such as deforestation from agricultural expansion) are very rare in boreal landscapes. We reclassified and condensed the original 23 classes into 14 land cover types (Supplementary Table 3) and determined the most dominant class for each sample plot. We used 6 main vegetation types in our analyses (see Study area and sampling design).

**Climate data.** Climatic variables were derived from the ERA5 monthly average climate data set with a spatial resolution of 0.25 decimal degrees[43] (Supplementary Table 2). We extracted data on monthly temperatures and precipitation for the past 40 years (1980–2019). We then calculated the annual mean metrics of these variables. We also quantified trends of annual temperatures and annual precipitations following the same approach as for tree cover change estimation. We considered climatic trends across two periods: One over the full-time range 1980–2019 and one which cover the period of tree cover change 2000–2019. Both periods are potentially relevant for changes in tree cover across the boreal biome. The long-term trend covers a period of time in which climatic changes became increasingly more observable in high latitudes and may have therefore influenced biome processes resulting in potential long-term effects on tree cover changes. The more recent time period is likely to have directly impacted tree cover changes within our study period.

**Elevation data.** We extracted elevation data from the USGS Global Terrain Elevation model with a spatial resolution of 250 m[94] (Supplementary Table 2).

## Data analysis
**Data extraction.** We used the ArcPro geographical information software (version 2.8.3) for generating the random sample plots and for extracting raster data to our sample plots.

**Spatial analyses.** The location of transects was informed by spatial autocorrelation of MODIS tree cover changes. We first created 10,000 random points across our study area and extracted MODIS-derived tree cover changes. We then tested for spatial autocorrelation of tree cover changes and described the best fit for the spatial variogram function by building linear mixed-effects models for the null model and a Gaussian, exponential, and spherical variogram function. We found that spatial autocorrelation was present (the null model performed worse than the spatially corrected models) and that an exponential function described this relation best. We created the spatial variogram by calculating variances in tree cover changes between points of equal distance from each other. We fitted an exponential variogram function to the data and identified the range, i.e., the distance at which variances level off (Supplementary Fig. 15). Transects were randomly selected while adhering to this minimum distance of 133 km.

**Standardisation of transect locations.** We compared mean tree cover changes of sample plots along all transect locations. Due to the geographic position of the North American boreal biome, we did not deem a definition of transect locations by latitude the best variable. That is because some latitudes are associated with both the northern and southern boundaries of the boreal. In these parts, the identification of potential patterns in tree cover changes would be confounded. We, therefore, standardised transect locations for each sample plot by considering different transect lengths in the following way: We assigned values of −1 and 1 to the intersections of each transect with the southernmost and northernmost boreal boundary, respectively. We then calculated the position of each plot along a transect within the −1 to 1 range based on the total number of plots between the south and north. This means that a plot at a location of 0 is halfway between the southern and northern boundary. All plots extending beyond boreal boundaries were assigned values up to −1.5 and 1.5 for the south and north, respectively. This way, irrespective of latitude or transect length, plots are characterised by their relative distance to each boundary. We use the term Standardised boundary distance (SBD) to refer to this distance.

Statistical analysis of tree cover changes and environmental conditions. We assessed the relationship between tree cover changes and environmental conditions through a general additive mixed-effects model (GAMM) using the mgcv package version 1.8-40 in R[95]. GAMMs can deal with the non-linearity that we expected between tree cover change and explanatory variables. We considered the following explanatory variables: SBD, mean tree cover 2000–2019, elevation, mean annual temperature, mean annual precipitation, changes in mean annual temperature, and changes in mean annual precipitation (each for the periods 1980–2019 and 2000–2019).

Prior to model fitting, we performed a principal component analysis using the stats package version 4.2.0 in R[96] to explore correlations between tree cover change and explanatory variables. We found high correlations between most explanatory variables (Supplementary Fig. 8). We, therefore, built models for each explanatory variable separately and performed model fitting with absolute and relative tree cover changes as separate response variables. We defined explanatory variables as fixed smooth terms and included interactions with all five disturbance types within the smoothing term to evaluate how the relationship between tree cover change and explanatory variables is modified by different disturbances. We treated individual transects as random effects. As the spatial analysis revealed spatial autocorrelation within transects, we included an exponential spatial correlation structure in the model. We validated all models visually using observed values, fitted values, and model residuals to check for heterogeneity and normality of residuals, patterns in model fits vs observed data, and any remaining random effects (see R script for details[97]).

We followed the same modelling process to quantify the relationship between tree cover change and vegetation types under different disturbances. We were interested in how tree covers changes with respect to the biome boundary for forests embedded in different vegetation types. We, therefore, included the interaction of all 30 combinations of vegetation types ($n = 6$) and disturbance types ($n = 5$) within the smoothing term of standardized boundary distance and fitted this model to the observed tree cover changes. All GAMM model results are listed in Supplementary Data 2.

All analyses described above were performed in R (version 4.2.0) using RStudio Version 1.1.463[98]. The accompanying R script is provided online[97].

### Reporting summary

Further information on research design is available in the Nature Portfolio Reporting Summary linked to this article.

## Data availability

Tree cover data extracted from MODIS VCF, Version 6 is available via the Application for Extracting and Exploring Analysis Ready Samples (AppEEARS). An updated CanLaD disturbance dataset on wildfire and timber harvest from 1985 to 2020 was kindly provided by Dr. Luc Guindon and will soon be available on the Government of Canada website. ERA5 climatic data on surface temperatures and precipitation are available on the Copernicus website, https://doi.org/10.24381/cds.f17050d7. Elevation data is available on the USGS website (https://doi.org/10.5066/F7J38R2N) following the link to the GMTED2010 Viewer. The global land cover map is available on Zenodo, https://doi.org/10.5281/zenodo.3243508. Boreal forest boundary data is available upon request to Sylvie Gauthier and Dominique Boucher. The map of tree cover trends produced by this study is available on an Zenodo, https://doi.org/10.5281/zenodo.7520322[99]. All source data are provided with this paper (Supplementary Data 1).

## Code availability

All codes used for the analyses are publicly archived on the lead author's GitHub[97].

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

## Acknowledgements

We express our deepest gratitude to Logan Berner for discussing our results and for putting them in perspective to his work on greenness indices, Sylvie Gauthier and Dominique Boucher for providing data on boreal forest biome boundaries, and Luc Guindon for an updated disturbance dataset for Canada. This work was carried out under the program of the Netherlands Earth System Science Centre (NESSC, grant number T5-WUR-MS-PhD1 awarded to M.S.), financially supported by the Dutch Ministry of Education, Culture and Science (OCW), with additional support from the Research Council of Norway (grant number 301922).

## Author contributions

R.R., M.H., M.S., E.v.N., J.U.J., and O.P.L.V. created and discussed the idea and study design. R.R. and C.X. were involved in data collection. R.R. performed data analyses. All authors discussed the results. R.R. wrote the manuscript. All authors revised and edited the manuscript.

## Competing interests

The authors declare no competing interests.
