## [Peer Review File · Nature Communications]

Review comments, first round

Reviewer #1 (Remarks to the Author):

Rothbarth and colleagues examined whether tree cover changes during recent decades were consistent with a northward shift of the Canadian boreal forest. Specifically, they examined changes in tree cover from 2000 to 2019 and assessed how climate and disturbance contributed to these observed changes. The authors quantified tree cover changes using MODIS satellite data and relied on existing forest disturbance and climate datasets. The analysis showed tree cover slowly increased across much of the northern interior boreal forest and tended to decline near the southern boundary. Fires and logging exacerbated tree cover loss near the southern boundary. This analysis contributes to growing evidence that the boreal forest may be shifting northward as the climate continues changing.

This is an important and timely topic given the potential for cascading impacts of a biome shift on ecological and social systems. The authors rely on a widely used tree cover data set and take a novel approach to characterize the relative proximity of each location to the northern and southern boundaries of the boreal biome, relying on what they call the “standardized boundary distance.” They also investigate factors contributing to observe tree cover changes using a rigorous statistical analysis accounts for spatial autocorrelation. There are strong elements to the analysis and paper, though I also have misgivings as detailed below.

Primary comments

As stated in the title, the authors primary conclusion is that “tree cover changes reveal contraction of North American boreal forests.” While the observed changes in tree cover are broadly consistent with a biome shift, it is not clear from the current analysis that there was a contraction of the boreal biome in recent decades. From my perspective, to conclude there was a contraction of the boreal biome, the authors would need to show that the total surface area covered by boreal forest recently decreased. Based on my reading of the analysis, that could potentially be the case, but has not been quantitatively demonstrated. It is also challenging to square the ideas that the boreal biome contracted but that “as a whole... forests in the North American boreal biome gained tree cover.”

How much of the observed increases in tree cover simply reflect vegetation recovery after historical fires and harvest rather than forest expansion associated with climate change? Currently, the analysis incorporates disturbance data from 2000 to 2015/2019 but does not try to account for potential impacts of historical disturbance. This makes it challenging to interpret the results simply in the context of a biome shift and not secondary succession.

The analysis includes tree cover changes in areas that were burned or logged and seems to counter-intuitively show that fires and logging increased tree cover in the northern boreal interior. This result needs to be presented and interpreted carefully. It’s hard to ecologically understand how areas that were burned (2000-2019) or harvested (2000-2015) would show an increase in tree cover from 2000-2019 (Figure 3), except for if the disturbances occurred early in the time series and these patterns reflect post-disturbance vegetation recovery. These disturbances did not in and other themselves increase in tree cover, but rather they killed off a portion of existing trees and the detected change reflects vegetation recovery after disturbance. Also, a significant portion of the perceived increase in tree cover after these disturbances could be related to deciduous shrubs

rather than trees. If a disturbance occurred in the early 2000s, then there would be 15-20 years of vegetation recovery that would likely be a mixture of trees and especially shrubs. If a disturbance occurred in the 2010s onward, then it's hard to see how the tree cover time series could possibly recover to and exceed pre-fire tree cover by 2019.

The paper purports to focus on the entire North American boreal forest; however, it appears the Alaskan boreal forest is not included in the analysis (Figure S9). Alaska is an important part of the North American boreal forest, so if the authors wish to provide a "biome-scale" analysis and couch the results in that context, then Alaska should be included. Focusing on the Canadian boreal forest is fine, but the wording should be adjusted to make it clear that is the focus rather than the whole North American boreal forest.

The results section describes what is shown in the figures but does not include any numerical summaries. Inclusion of numerical summaries would help to better convey the magnitude of tree cover change and how it varies across the biome. For instance, the authors state "As a whole, however, forests in North American boreal biome gained tree cover." This leads me to wonder how much on average did tree cover increase across the study domain? The authors also note that "tree cover gains were strongest in the northern half of the boreal interior. At and beyond the current northern boundary, tree cover changes were low." Again, statements like this benefit from numerical summaries that illustrate the magnitude of changes. Overall, this section would be much stronger and more compelling if it included numerical summaries supporting the primary statements.

The Canadian boreal forest is huge. Tree cover changes are unlikely to be homogenous across either the northern or southern boundaries of the biome. The paper would be strengthened by reporting on and discussing some of the regional variability in dynamics across these boundaries, which would also make it possible to compare results from this analysis with regional field studies.

Line specific comments

Line 19: The authors probably mean to say "contraction at the southern and expansion at the northern boundary" rather than "expansion at the southern and contraction at the northern boundary."

Line 32: The authors state, "Boreal forests are the largest terrestrial biome on earth..." but I don't believe this is accurate. Of course, there are different biome definitions, but most sources put the tropical forest as the largest forest biome (e.g., Pan et al., 2011; Sexton et al., 2016).

Line 33: Need a reference for the statement that the boreal forest acts as "... a critical stabilizer of the climate system."

Line 34: Need references for the statement that "the integrity of the boreal biome is increasingly undermined by climate change."

Line 44: Change "prolonged" to "longer"

Line 45: the sentence suggests that

Line 46: Delete "... increases in high-latitude" because it's redundant with other parts of the sentence.

Line 47: Consider citing some newer review studies such as Rees et al. (2020) and Mekonnen et al. (2021) instead of Harsch et al. (2009) and Myers-Smith et al. (2011), respectively.

Line 49: Add citations supporting the statement that “growing conditions of cold adapted boreal tree species may become too hot and dry.”

Line 53: Does the “northward migration of boreal woody species” mean trees and shrubs? If so, then the second portion of the sentence does not make sense, but if not then just say “trees”.

Line 58: What does “deteriorating growing conditions” mean? Hotter and drier?

Line 59: “reduction” ◊ “contraction”

Line 61: Here and elsewhere, consider using the term “transient states” instead of “transients”

Line 81: Provide a short justification for why “We expect needleleaf forests in the south to sustain greater tree cover losses than broadleaf mixed leaf forest.”

Line 139: The authors state “In the absence of fire and harvest, boreal tree cover increased across most of the boreal distribution range. This increase was lower in the south than the north.” Please provide a percentage rather than saying “most” and provide numerical summaries for the south versus north.

Line 232: Northern expansion could also be limited by cold, nutrient poor soils (Ellison et al., 2019; Sullivan et al., 2015) and herbivores (Olnes et al., 2017).

Line 248: For this subsection, consider comparing with results from Wulder et al. (2020).

Line 258: The authors state, “These observations suggest a lower resilience of southern boreal forests to wildfire perturbations and the faster proposed by recovery of northern interior forest.” These are both substantial conclusions to draw from the current analysis, though I’m not sure the analysis was designed to directly test these ideas. If the authors want to make these statements, then I would encourage them to dig more deeply into the literature around these topics and cite that literature herein.

Line 273: The authors state, “Tree cover changes are strongly linked to climate.” However, the strength of this relationship is not clearly demonstrated in the results. How much of the variability in tree cover changes was explained by climate when using the GAMMs? Consider adding this information to the results.

Line 280: But also see Girardin et al. (2016).

Line 290: Shrubs have certainly expanded in the Arctic, but there is limited evidence for actual northward shrub expansion (Myers-Smith & Hik, 2018).

Line 296: ... “in combination with losses related to higher rates of warming...” This pattern isn't evident in Fig 5 which rather seems to suggest maybe even slightly increasing tree cover in areas

that warmed more rapidly.

Line 321: See also Koven (2013)

Line 325: This paragraph about albedo doesn't have a single reference...

Figure 2:

- It's important to show the magnitude tree cover change (%) instead of just gain and loss.
- Panel C is missing its panel letter
- Could the perceived decrease in tree cover beyond tree line be due to noise in the tree cover data set? The MODIS satellites have experienced sensor degradation and orbital changes over their few decades of operation that influence measurements of surface reflectance, spectral indices, and almost certainly derived products like tree cover. If tree cover is very low (e.g., say 2%), then it would only take a very small amount of error in tree cover predictions to cause the perceived decline (e.g., say 1%).
- The legend key is obstructing the x-axis
- The text size is very small throughout the figure. Consider making all bigger.
- Do the black lines in panels a and b convey the average change that was then smoothed using a running mean?
- The standard error bands should be symmetrical around the black line, correct? However, across much of the figure the error band appears obstructed by the solid polygon.

Figure 4. Spell out SBD and TC in the axes.

Ellison, S. B., Sullivan, P. F., Cahoon, S. M., & Hewitt, R. E. (2019). Poor nutrition as a potential cause of divergent tree growth near the Arctic treeline in northern Alaska. *Ecology*, 100(12), e02878.

Girardin, M. P., Bouriaud, O., Hogg, E. H., Kurz, W., Zimmermann, N. E., Metsaranta, J. M., . . . Büntgen, U. (2016). No growth stimulation of Canada's boreal forest under half-century of combined warming and CO₂ fertilization. *Proceedings of the National Academy of Sciences*, 113(52), E8406-E8414.

Harsch, M. A., Hulme, P. E., McGlone, M. S., & Duncan, R. P. (2009). Are treelines advancing? A global meta-analysis of treeline response to climate warming. *Ecology letters*, 12(10), 1040-1049.

Koven, C. D. (2013). Boreal carbon loss due to poleward shift in low-carbon ecosystems. *Nature Geoscience*, 6, 452-456. doi:10.1038/ngeo1801

Mekonnen, Z. A., Riley, W. J., Berner, L. T., Bouskill, N. J., Torn, M. S., Iwahana, G., . . . Grant, R. F. (2021). Arctic tundra shrubification: a review of mechanisms and impacts on ecosystem carbon balance. *Environmental Research Letters*, 16(5), 053001.

Myers-Smith, I. H., Forbes, B. C., Wilmking, M., Hallinger, M., Lantz, T., Blok, D., . . . Lévesque, E. (2011). Shrub expansion in tundra ecosystems: dynamics, impacts and research priorities. *Environmental Research Letters*, 6(4), 045509.

Myers-Smith, I. H., & Hik, D. S. (2018). Climate warming as a driver of tundra shrubline advance. *Journal of Ecology*, 106(2), 547-560.

Olnes, J., Kielland, K., Juday, G. P., Mann, D. H., Genet, H., & Ruess, R. W. (2017). Can snowshoe hares control treeline expansions? *Ecology*.

Pan, Y., Birdsey, R. A., Fang, J., Houghton, R., Kauppi, P. E., Kurz, W. A., . . . Canadell, J. G. (2011). A large and persistent carbon sink in the world's forests. *Science*, 333(6045), 988-993.

doi:10.1126/science.1201609

Rees, W. G., Hofgaard, A., Boudreau, S., Cairns, D. M., Harper, K., Mamet, S., . . . Tutubalina, O. (2020). Is subarctic forest advance able to keep pace with climate change? *Global Change Biology*, 26, 3965–3977. doi:10.1111/gcb.15113

Sexton, J. O., Noojipady, P., Song, X.-P., Feng, M., Song, D.-X., Kim, D.-H., . . . Pimm, S. L. (2016). Conservation policy and the measurement of forests. *Nature Climate Change*, 6(2), 192-196.

Sullivan, P. F., Ellison, S. B. Z., McNown, R. W., Brownlee, A. H., & Sveinbjörnsson, B. (2015). Evidence of soil nutrient availability as the proximate constraint on growth of treeline trees in northwest Alaska. *Ecology*, 96(3), 716-727. doi:10.1890/14-0626.1

Wulder, M. A., Hermosilla, T., White, J. C., & Coops, N. C. (2020). Biomass status and dynamics over Canada's forests: Disentangling disturbed area from associated aboveground biomass consequences. *Environmental Research Letters*, 15(9), 094093.

Reviewer #2 (Remarks to the Author):

The authors utilize time series of tree cover and disturbance data to characterize changes in the distribution of boreal forests in Canada and identify two key findings: the southern boundary is experiencing faster tree cover decline than the northern boundary's tree cover is expanding, resulting in a net reduction in boreal forest coverage, and that the rates of tree cover changes are related to climate change, wildfires, and timber harvest. This is an interesting and significant question, with implications, as the authors point out, for the global carbon-climate system.

I am broadly supportive of the line of inquiry pursued by the study. This is a key question with lots of circumstantial or smaller scale evidence, and a continental-scale examination of boreal biogeography is becoming more feasible with recent advances in remote sensing and computational power. However, the study is marred by several crucial flaws related to its framing, methodology, and reporting of results, and I am not sure it is either accessible to a general audience or detailed enough to be credible.

Overall, I found that the paper was clearly written, but also found it lacking major holes in its discussion of the ecological processes and the relevant literature. I was surprised that the introduction had little description of the relatively deep literature regarding Arctic-boreal greening trends in North America. Many others have described how these positive trends in ecosystem productivity have been identified using time series of a unitless index, the normalized difference vegetation index, but one of the challenges here is translating the trends in an index to actual ecological change. To me one of the key potential advances of this paper is the examination of such trends using an ecologically/physically meaningful quantity, tree cover, which is closely related to (but not exactly the same as) relevant values like carbon storage, carbon flux, albedo, and more. I was disappointed that the authors seemed to gloss over this, beyond failing to provide to a general readership the basic framework for our understanding of boreal forests and global change. Similarly, several key ecological processes, such as the importance of changing boreal forest types or the likely large importance of post-disturbance recovery, are referenced in the results and somewhat in the discussion, but the general readership may be lost as to the relevance of these factors considering their lack of mention in the introduction. The research questions are great, but little is provided to place them in context for the reader beyond the somewhat overstated boreal forest changes. To

summarize, I recommend including at least one paragraph in the introduction and elaborating on the discussion to provide a more complete framework of boreal forests and global change.

A somewhat related point - I found that the way the authors wrote the paper was a bit sensational. For one, it rubbed me the wrong way that the abstract and title are referencing "North American boreal forests", but in reality the study focuses solely on Canada. Excluding Alaska is not necessarily a problem, but it does represent quite a large part of the boreal North America, and the paper implies it is representing boreal forests across North America. I suggest toning down scope of the title and abstract so readers are not misled.

Second, I had several issues with the methods used. We are living in an age where continental scale remote sensing with medium spatial resolution data over a relatively short time series is widely accessible through platforms like NASA's data portals and computational tools like Google Earth Engine. Why limit the analysis to the average over 10,000 plots covering 36 km² each? The boreal forest experiences very spatially heterogeneous, sporadic, and localized disturbances that are hard to characterize so simply with such large areas and spanning just 6% of the domain. This might be okay if the only process in question was something more broadly distributed, such as climate effects. But the large importance of disturbances, which are relatively sporadic and localized in nature, means that a relatively sparse sampling (covering, as the authors state, just 6% of the domain), could miss lots of the disturbance-driven tree cover losses and I am not convinced that the distribution is representative of the disturbance regimes. Especially considering the missing harvest data for 2016-2019. Because the disturbances play a key role in the boreal forest and in the study's results, I found the resulting analysis hard to believe. This is especially true given the exceedingly small rates of tree cover change (roughly 0.5% per year... or 10% over the study period, feels rather smaller magnitude than I'd expect), despite the occurrence of stand-replacing disturbances. The approach needlessly discards lots of information, when a fundamentally biogeographic question should be more geographically comprehensive and include some sort of area estimation. A more complete analysis would also bolster the climate modeling section - I found the GMM results a bit hard to follow since the relationships between tree cover change and climate looked rather unstable and perhaps overfit.

I was sad to see a lack of a map of the actual trends in tree cover change. It is an interesting data product that I believed the authors have generated, but it is hard to evaluate when it is never shown. Instead, we are able to see largely aggregated/averaged values over certain subdomains that discard much of the interesting geographic variability. Showing a binary 'gain/loss' map, especially without regards to statistical significance or magnitude, is a regrettable simplification of the changes in the boreal forest.

And finally, somewhat reiterating a point about the introduction - what about disturbances prior to 2000? There are many in this domain and recovery from these disturbances must be structuring a very large part of the tree cover growth in the boreal interior. I was surprised to see it described simply as "undisturbed" and still that it received relatively little coverage in both the introduction and the discussion. Some mention is made of forest resilience to these disturbances, but to me it is a much larger and more significant story than the relationship with climate. There is an interesting avenue to pursue by using the disturbance data, which goes back to 1985 rather than 2000, to characterize post-disturbance recovery more specifically.

And relatedly - why use a static land cover map? A key part of the analysis is stratifying and filtering

by land cover type, but a map for just the year 2015 was used. If disturbance and recovery are key parts of the interpretation of the results, I would expect land cover change to also be significant. How can you incorporate land cover change, which almost certainly occurs in boreal wildfires and timber harvest, into the story?

In summary, I believe the authors are pursuing an interesting research question that will be of interest to a potentially large audience of ecologists, global change biologists, and Earth system scientists, but that the study as presented requires a rework of its framing, a more complete method, and a better description of its results. In its current state it feels somewhat incomplete, and not an especially robust addition to a research question that has a lot of activity. It is not ready to be published in Nature Communications as it is now.

Some more line and figure specific comments follow:

Line 22: more specific to say "latitudinal" rather than "geographical"

Also line 22: I would put in parentheses the years covered after two decades (2000-2019)

Line 24: A bit confusing how this is written. I think it's clearer to say "Tree cover losses due to wildfire and timber logging accentuated the tree cover losses in undisturbed region..."

Line 25: is resilience really a part of this paper?

Line 27: Can you be more specific about the impacts of a boreal contraction? It's not that compelling to say "with implications for...".

The abstract is a bit vague and a bit weak. Including some of the key quantities that you calculated would be helpful.

Lines 32-36: The first few sentences of the introduction feel out of order. I would say that boreal climate change is twice the global average before saying that this climate change is threatening the integrity of the boreal biome.

Lines 32-41: This is a general audience, and as such I feel it is important to summarize briefly what are some of the implications for carbon and climate for changes in boreal forest extent. Why should the average reader care? For example, forest loss and disturbances are reducing the boreal carbon sink in North America (see the recent paper in Nature Climate Change, Wang et al 2021). It should be brief, but clear... there are plenty of angles for this context.

Lines 42-51: This paragraph is well-written. Good job! One comment - I would like to see a citation for the statement that the conditions in the south will become too hot and dry for boreal tree species. Would that lead to mortality? Or slower recruitment/establishment? It's a little bit vague, in contrast to the northward expansion part.

Line 52: remove the word 'differently'

Line 70: For a general audience, define MODIS . You might also mention here that it is time series data.

Line 71: Your study is really more about Canada than North America broadly. At this point I was expecting something broader and was confused later when only Canada was studied. Make it clearer here.

Lines 78-82: The hypothesis that land cover type is an important factor is very interesting, but was not set up. This will be confusing for non-specialists. Consider a brief discussion of why these forest

types are important earlier in the introduction.

Figure 1: The conceptual diagram looks nice, but I am not sure that it is necessary. It's quite busy, so I found it confusing at first. The core concepts were already well-explained in the text. It is also a bit unintuitive that the northward expansion is described in a left-right manner. But overall I would remove it.

Line 104: mean tree cover? I assume you mean for 2000? This is not described in the methods. Were tree cover changes calculated as just the difference between 2019 and 2000, or as a trend fit to time series?

Figure 2: Why not show the trends in tree cover change?? It is challenging to interpret the map because of the binary nature of it. Are there really no regions that are neutral or near neutral? For example, it is confusing to see losses recorded in the Mackenzie mountain range, which has very limited tree cover. It would make the importance of fires and other disturbances much clearer to see their relative impact on some of these values. Also - why not label the map as panel C? It was hard to find its description in the caption. It would also help to see the line indicating the "0" standardized boundary distance (e.g. exactly halfway between the boundaries) across the map.

Showing both panels A and B seemed unnecessary, especially because you state in the text that because the results are relatively similar you stick to absolute tree cover changes.

Last note about Figure 2: I recommend using a different projection for displaying areas in this part of the world. Using a Mercator projection greatly distorts areas at high latitudes, making this map difficult to interpret when we are comparing areas.

Lines 135-137: I do not think that you can claim much of the boreal forest is truly 'undisturbed'. You should qualify that these areas have not been disturbed since 2000 (as you state in Figure 3)

Figure 3: This was an informative figure. In Panel B, I found it very curious that tree cover should increase so much in the northern part of the domain for both burned and harvested forests. Why aren't any of the fire trends negative here, yet many negative values occur in the southern parts of the domain in Panel A?

Figure 4: Cool figure. I would recommend spelling out standardized boundary distance on the Y axis, since you have space.

Lines 188-189: Why analyze climate trends across 1980-2019? Isn't this extending beyond the limits of the tree cover data?

Line 229: Why should this contraction of forests be transient? It's not clear to me. If disturbances occurred frequently enough to cause some of this southern decline in 2000-2019, why shouldn't we expect disturbances to continue to reduce tree cover going forward? We have already seen major fires in Canada since 2019 - if anything, I might expect the contraction to accelerate.

Lines 237-246: Thank you for bringing up, albeit briefly, the literature on Arctic/boreal greening trends. I think this could be discussed further, both in the introduction (to give context) and here, to explain more why your results are novel. I see the novelty in using physical units to describe

greening trends.

Line 252: I don't know that the citation listed here, Sulla-Menashe et al, really supports this discussion of disturbances driving forest loss. Sulla-Menashe et al were more describing trends in greenness or productivity here, which is not necessarily the same thing as changes in forest cover or biomass. I would find something more directly relevant.

Lines 256-258: To evaluate these comparisons in wildfire regimes, it would be necessary to provide some sense of the distribution of fires that occurred in your domain going back further than 2000. I believe that a number of major fires occurred in the 1990s, for example, in the interior of western Canada, while the period from 2000-2013 had relatively lower rates of fire. The increased interior tree cover may be more a matter of sampling than a real signal of different forest resilience to disturbance. Overall, there seems to be little discussion of post-fire recovery, but I think it is in fact dominating much of the signal in this study. I think there could be an interesting analysis using the CANLaD data to attribute some of these trends to post-fire recovery.

Lines 304-305: This statement feels rather speculative, especially without any citations to back up this notion or elaboration on why this type conversion could happen. I would remove it or expand it, but not leave it as it is.

Lines 312-324: Much is said here about the losses in tree cover potentially resulting in carbon source. But the total tree cover increased in the domain, according to your data, which makes me confused why we are discussing the possibility of a carbon source. Even if it is just small areas that become a source, it seems to me the bigger story is the total increase in tree cover. This doesn't flow from your results, or I am missing something.

Also, isn't the majority of the carbon in the boreal biome located in the thick permafrost soils? If we are discussing carbon budgets in Canada, I do not think this can be ignored.

Lines 325-331: Plenty of research discussing changes in albedo in boreal forests that is missing here.

Lines 334-338: Regarding MODIS limitations, what about new satellite datasets? Such as new lidar or higher resolution imagery?

Line 321-322: There is lots of research investigating disturbance related carbon losses vs warming induced carbon gains in Canada. I feel that this section is missing a lot of discussion from the literature. It should be expanded to provide a more nuanced view of the problem.

Line 374: I am a little confused as to why the data were analyzed in these sample plots. Why not analyze the entirety of the dataset? Especially considering you are addressing a fundamentally biogeographic question that is impacted by sporadic, localized disturbances, I would think geographic completeness is crucial. Certainly, many other studies have analyzed much larger datasets (e.g. at higher resolution and geographically complete, over longer timescales), so I am not sure that computational limits should explain this.

Line 384: I am surprised that you condense each plot into essentially a single value. That is quite a lot of information being lost, especially since many disturbances and climatic gradients here occur at smaller scales than 36 km².

Line 401: What mean tree cover? In 2000, or over 2000-2019?

Line 403: You are not calculating biomass change here. Biomass and cover, while sometimes correlated, are quite different values.

Lines 418-420: I looked at Figure S11, and I'm not sure how you can use this to be confident that not having harvest data for 20% or so of your study period does not change trends. Especially considering your sampling approach, which might not be representative of the overall geographic patterns of disturbance.

Lines 422-424: I do not think you should call the remaining plots "undisturbed". They have not been burned or harvested since the year 2000, but most of the forest in this domain will likely have been disturbed relatively recently before that. You can investigate this with the CanLaD dataset, which includes observations to 1985. It would at least help us understand a little more about post-disturbance recovery.

Lines 435-440: Why would the climate change in the time period before your tree cover data have relevance to the changes in tree cover observed? I do not follow why the 1980-2019 ERA5 data were used.

Lines 456-468: I liked the approach for developing the SBD.

Reviewer #3 (Remarks to the Author):

Generally, the study provides important contribution to the understanding of the interacting influences of climate change and disturbances to the envisaged northward shift in the spatial coverage of the boreal biome. It adds to this understanding, a continental scale evidence on percent tree cover changes along a known south-north gradient of decreasing tree cover. Further, tree cover changes along this gradient are assessed to show the influence of timber harvest and fire in different vegetation cover types, thus providing rich information on some of the main interacting factors contributing to the tree cover dynamics in the biome.

Suggested improvements:

1. The first line of the abstract reading "Climate change is expected to shift the boreal biome northward through expansion at the southern and contraction at the northern boundary respectively." seems to suggest '... expansion at the northern (not southern as presented) and contraction at the southern (not northern as presented) ...' unless the direction of expansion/contraction is specified in the current sentence. Please check and correct as may be relevant.
2. The two decades duration of the study needs to be explained in terms of its potential and/or limitation in the context of the aims of the study and the influence of disturbances. e.g. how fast are the regeneration times within the biome? would one expect different tree cover trends and thus different conclusions if the MODIS tree cover data set was available for the past 40yrs or 70yrs or

so?

3. The two sentences presented in Lines 141 - 144, namely "Losses of tree cover in disturbed areas were highest at the southern boundary where wildfires and harvests overlap in extent (Figure S3). Interestingly, tree cover in northern boreal forests increased, despite the occurrence and greater extent of wildfires in these parts compared to the southern boreal (Figure S3).", provide information on results presented in both Figure 3 and Figure S3 but only cite Figure S3. To improve on clarity, either the two sentences can be rephrased to strongly link them with the preceding sentence that cites Figure 3, or Figure 3 can be cited again in these two sentences.

4. The word "scale" in the sentences "We found that the scale of this absolute change depended on mean tree cover across our study period" (Line 399 to 400) and " ... while relative values are looking at the scale of change with respect to initial conditions." (Line 404). Does it refer to the temporal scale? or spatial scale? for the whole study or scale in the sense of size/magnitude of change for each sample plot? Please clarify as may be necessary.

5. The text suggests a high confidence that missing timber harvest data for the period 2016-2019 does not considerably change tree cover trends (Lines 418 - 420). This is further elaborated in Figure S11. This figure, particularly panels A and B suggest "Tree cover trends in the south were generally higher for the longer time period than the shorter one" (lines 119 - 120 of the supporting file). However, the longer period (20 years) is just about four years longer than the shorter period (16 years), which may imply that the four years are contributing some interesting information in the running means such that higher trends are observed over the longer period. Was the tree cover data for the period 2016-2019 examined independently? or relative to that of the period 2000-2015 (where timber harvest data is available) or that of 2000-2019 (the longer period)? Although results in panels C and D justify further analyses without the missing timber harvest data, it may be interesting to further explain the tree cover data/trends/changes for this 2016 - 2019 period, particularly because timber harvesting may directly influence: i.the detected annual tree cover, and trends in tree cover as evident in Figure 3B and ii. arguably, the successive years' fire behavior.

6. The acronym "SBD" is explained to refer to "Standardised boundary distance" (Line 468 of the main manuscript document). However, in some cases "standardized boundary position" is used (e.g. Figures S1, S2, S11, etc.). This raises a question as to whether "Standardised boundary distance" and "standardized boundary position" are synonymous or they are meant to refer to different characteristics of the sample plots, as in some cases both are utilized next to each other (e.g. Figure S11). The use of the two phrases needs to be clarified and the presentation needs to be consistent both in the main manuscript and the supplementary material document. It is further suggested that, whenever the acronym "SBD" is utilized to label figure axes (e.g. the y-axis of Figure 4, panel A of Figure S5, Figure S7, panel C of Figure S11, etc.), it can be written in full either on respective figure's axis or in its caption.

7. The figure label "S4 (B)" is repeated twice while "S4 (C)" is missing. Please correct.

8. Future studies may be suggested to look further into changes in species composition within the land cover types along the south-north gradient, particularly in areas with notable tree cover changes. The current findings illustrate that timber harvests occur heavily in the south. This may possibly be explained by higher tree cover in the south, or availability of preferred timber species, etc. In the presence of fire and other factors (both natural and human induces), different species

within the cover types may exhibit different levels in their regeneration rates, colonization rates, resilience or sensitivity to disturbances, etc., which in turn may influence the disturbance patterns and may provide feedback to the transient tree cover trends at a local- or biome-level scale. Availability of more detailed information on any aspect of these dynamics may further inform management plans and policies aiming at minimizing carbon sources.

**Response to reviewers**

**Reviewer #1 (Remarks to the Author):**

Rotbarth and colleagues examined whether tree cover changes during recent decades were consistent with
a northward shift of the Canadian boreal forest. Specifically, they examined changes in tree cover from
2000 to 2019 and assessed how climate and disturbance contributed to these observed changes. The e
authors quantified tree cover changes using MODIS satellite data and relied on existing forest disturbance
and climate datasets. The analysis showed tree cover slowly increased across much of the northern interior
boreal forest and tended to decline near the southern boundary. Fires and logging exacerbated tree cover
loss near the southern boundary. This analysis contributes to growing evidence that the boreal forest may
be shifting northward as the climate continues changing.

This is an important and timely topic given the potential for cascading impacts of a biome shift on ecological
and social systems. The authors rely on a widely used tree cover data set and take a novel approach to
characterize the relative proximity of each location to the northern and southern boundaries of the boreal
biome, relying on what they call the "standardized boundary distance." They also investigate factors
contributing to observe tree cover changes using a rigorous statistical analysis accounts for spatial
autocorrelation. There are strong elements to the analysis and paper, though I also have misgivings as
detailed below.

>> We are grateful for the kind words of reviewer 1 in regards to the novelty and importance
of our study and the robustness of our chosen analyses.

**Primary comments**

As stated in the title, the authors primary conclusion is that "tree cover changes reveal contraction of North
American boreal forests." While the observed changes in tree cover are broadly consistent with a biome
shift, it is not clear from the current analysis that there was a contraction of the boreal biome in recent
decades. From my perspective, to conclude there was a contraction of the boreal biome, the authors would
need to show that the total surface area covered by boreal forest recently decreased. Based on my reading
of the analysis, that could potentially be the case, but has not been quantitatively demonstrated. It is also
challenging to square the ideas that the boreal biome contracted but that "as a whole... forests in the North
American boreal biome gained tree cover."

>> We agree with reviewer 1 that our results are consistent with a biome shift based on tree
cover changes. Reviewer 1 further mentions that a demonstration of surface area change across
the boreal biome is needed to conclude that a contraction may be underway. As tree cover is a
percentage of treed area, it is also a measure of surface area. Our observation of tree cover
increases across the biome therefore means that the total surface area of boreal forests has
increased. We confirmed that the magnitude of this increase based on our sample plots (mean
of 0.12% per year) is comparable to the increase across the entire biome using the map in
Figure 2 in the manuscript (mean of 0.1% per year). As discussed in our manuscript, this area
gain is mainly driven by gains in the northern interior. Purely based on areas change as
suggested by reviewer 1, the boreal biome did not contract. However, the focus of our analyses
centred around a biome shift and around a possible contraction in geographical extent of tree
cover distribution. The observed tree cover losses at the southern extent were not compensated
by a geographical expansion of tree cover at and beyond the northern boundary. We interpret
this mismatch of tree cover change as potential for a contraction of the biome extent.
Consequently, while the extent of the biome may contract, tree cover (and thus area) may
increase. Both definitions of a biome shift/contraction (i.e. geographical extent and area
change) are important and are rather complementary than contradictory. We have included a
both forms of contraction in our results (lines 160-163) and discussion (lines 317-321). We
have also added a brief section in the supplementary materials that demonstrates the

simultaneous occurrence of a contraction of the biome distribution range and an increase in
surface area (Figure S2).

How much of the observed increases in tree cover simply reflect vegetation recovery after historical fires
and harvest rather than forest expansion associated with climate change? Currently, the analysis
incorporates disturbance data from 2000 to 2015/2019 but does not try to account for potential impacts
of historical disturbance. This makes it challenging to interpret the results simply in the context of a biome
shift and not secondary succession.

>> We thank reviewer 1 for the idea to include disturbance data between 1985 and 2000 from
the CanLaD data set which is consistent with comments by reviewer 2 about the same aspect.
Originally, we decided against such an inclusion, as we intended to analyse the potential
reasons for decreasing tree cover in the period 2000-2019 (limited by the MODIS availability)
However we agree that it is also interesting to study the longer term recovery of areas that
have been disturbed before 2000. We have therefore followed the reviewers' suggestion and
included a separation of historical and more recent disturbance data through the following
categories: (1) Undisturbed since 1985, (2) Wildfire 1985-2000, (3) Timber harvest 1985-2000,
(4) Wildfire 2000-2019 and (5) Timber harvest 2000-2019. This separation now forms part of
the presentation of results (mainly lines 200-226) and the discussion (lines 344-396) and has
greatly improved the analysis.

The analysis includes tree cover changes in areas that were burned or logged and seems to counter-
intuitively show that fires and logging increased tree cover in the northern boreal interior. This result needs
to be presented and interpreted carefully. It's hard to ecologically understand how areas that were burned
(2000-2019) or harvested (2000-2015) would show an increase in tree cover from 2000-2019 (Figure 3),
except for if the disturbances occurred early in the time series and these patterns reflect post-disturbance
vegetation recovery. These disturbances did not in and other themselves increase in tree cover, but rather
they killed off a portion of existing trees and the detected change reflects vegetation recovery after
disturbance. Also, a significant portion of the perceived increase in tree cover after these disturbances
could be related to deciduous shrubs rather than trees. If a disturbance occurred in the early 2000s, then
there would be 15-20 years of vegetation recovery that would likely be a mixture of trees and especially
shrubs. If a disturbance occurred in the 2010s onward, then it's hard to see how the tree cover time series
could possibly recover to and exceed pre-fire tree cover by 2019.

>> We appreciate the valuable contribution of reviewer 1 to the interpretation of the indeed
remarkable results of tree cover increases in the northern boreal under the presence of
disturbances. We agree with the reviewer that the change in tree cover is only interpretable in
the context of disturbance recovery. This interpretation already formed part of our discussion
in which we also include the very important suggestion of the reviewer regarding the
contribution of deciduous species to tree cover increases (lines 367-370). However, a possible
contribution of shrubs to tree cover changes (as indicated by reviewer 1) is, in our opinion,
unlikely, as the MODIS VCF data excludes vegetation below 5m in height from the estimation of
tree cover and considers shrubs under the component named 'Non-tree vegetation' (see details
in the user guide¹).

Finally, reviewer 1 rightly points to the importance of disturbance timing for the assessment of
tree cover trends. We agree that recent disturbances have a more negative effect on tree cover
trends than disturbances further back in time. However, we do not share the conclusion that
recently disturbed areas are unlikely to show positive trends and are ecologically difficult to
justify. We found that trends in recently disturbed areas can be positive under the following
three scenarios or a mix thereof: (1) disturbances did not lead to complete stand replacement
in our sample plots, (2) pre-disturbance trends were highly positive so that disturbances had a

smaller impact, (3) post-disturbance recovery was high. Especially scenario 1 is quite common,
as the proportion of our sample plots affected by disturbances is usually small (Figure S5),
resulting in tree cover loss but not complete replacement. In those instances, positive tree cover
trends in unaffected areas within sample plots can compensate for disturbance losses, leading
to positive tree cover trends.

While positive tree cover trends occur both in northern and southern boreal forests, they are
far more dominant in the northern interior. This may be a result of improved growing conditions
in the north which facilitate widespread and fast recovery. Both mathematically and
ecologically, positive tree cover trends in disturbed areas are therefore possible.

The paper purports to focus on the entire North American boreal forest; however, it appears the Alaskan
boreal forest is not included in the analysis (Figure S9). Alaska is an important part of the North American
boreal forest, so if the authors wish to provide a "biome-scale" analysis and couch the results in that
context, then Alaska should be included. Focusing on the Canadian boreal forest is fine, but the wording
should be adjusted to make it clear that is the focus rather than the whole North American boreal forest.

>> Both reviewer 1 and reviewer 2 raise an important point. Alaska indeed forms a large part
of the North American boreal forest biome. Data sets on disturbances for Alaska, however, differ
in approach, scale and methodology and thus limit the generalisation of results and implications
across the entire North American biome. To match the very valid point reviewer 1 and 2 make,
we have now extended our tree cover trend analysis to include Alaska and report results across
the North American boreal biome. We then focus the analysis of drivers of tree cover trends
(i.e. disturbance, climate etc.) on Canadian boreal forests as already done in the former version
of the manuscript.

The results section describes what is shown in the figures but does not include any numerical summaries.
Inclusion of numerical summaries would help to better convey the magnitude of tree cover change and
how it varies across the biome. For instance, the authors state "As a whole, however, forests in North
American boreal biome gained tree cover." This leads me to wonder how much on average did tree cover
increase across the study domain? The authors also note that "tree cover gains were strongest in the
northern half of the boreal interior. At and beyond the current northern boundary, tree cover changes were
low." Again, statements like this benefit from numerical summaries that illustrate the magnitude of
changes. Overall, this section would be much stronger and more compelling if it included numerical
summaries supporting the primary statements.

>> We have followed the suggestion by reviewer 1 to include numerical summaries in the
results section.

The Canadian boreal forest is huge. Tree cover changes are unlikely to be homogenous across either the
northern or southern boundaries of the biome. The paper would be strengthened by reporting on and
discussing some of the regional variability in dynamics across these boundaries, which would also make
it possible to compare results from this analysis with regional field studies.

>> We agree with reviewer 1 and now report on regional differences in tree cover changes. To
aid in reporting in a structured manner, we use Canadian ecozones. The Canadian Council on
Ecological Areas map of ecozones is a commonly used map to report such results and we have
now summarised results on tree cover trends for ecozones in the results section and as maps
in the supplementary information. In addition, we now present a map showing tree cover
changes across the North American boreal biome.

Line specific comments

Line 19: The authors probably mean to say "contraction at the southern and expansion at the northern
boundary" rather than "expansion at the southern and contraction at the northern boundary."

>> We have corrected this in the text (lines 19/20).

Line 32: The authors state, "Boreal forests are the largest terrestrial biome on earth...," but I don't believe
this is accurate. Of course, there are different biome definitions, but most sources put the tropical forest
as the largest forest biome (e.g., Pan et al., 2011; Sexton et al., 2016).

>> We changed this to 'one of the largest terrestrial biomes on Earth' to reflect differences in
definition (line 34).

Line 33: Need a reference for the statement that the boreal forest acts as "... a critical stabilizer of the
climate system."

>> We changed this to 'critical tipping element of the climate system' and cited Lenton *et al.*
(2008), PNAS (line 35/36).

Line 34: Need references for the statement that "the integrity of the boreal biome is increasingly
undermined by climate change."

>> We cited Gauthier *et al* (2015), Science (line 37).

Line 44: Change "prolonged" to "longer"

>> Changed accordingly (line 47).

Line 45: the sentence suggests that

>> This comment is incomplete.

Line 46: Delete "... increases in high-latitude" because it's redundant with other parts of the sentence.

>> Deleted accordingly (line 49).

Line 47: Consider citing some newer review studies such as Rees et al. (2020) and Mekonnen et al. (2021)
instead of Harsch et al. (2009) and Myers-Smith et al. (2011), respectively.

>> We added the suggested references but kept the older ones, as the latter are important
early contributions (line 50).

Line 49: Add citations supporting the statement that "growing conditions of cold adapted boreal tree
species may become too hot and dry."

>> We added D'Orangeville *et al* (2018). We also included recent publications on shifts from
boreal to temperate species in line 54-55.

Line 53: Does the "northward migration of boreal woody species" mean trees and shrubs? If so, then the
second portion of the sentence does not make sense, but if not then just say "trees".

>> Changed to 'trees' (line 65).

Line 58: What does "deteriorating growing conditions" mean? Hotter and drier?

>> Changed to '...hotter and drier growing conditions...' (lines 69-70).

Line 59: "reduction" □ "contraction"

>> This comment is not entirely clear but we have changed the word 'reduction' with
'contraction' (line 72).

Line 61: Here and elsewhere, consider using the term "transient states" instead of "transients"

>> We believe that, contextually, 'transients' is the more appropriate term, as we discuss the
prospect of transient processes, not transient states per se. Transient states might suggest
constant conditions, while we stress the transient trajectories. The boreal biome contraction is
one such trajectory which is transient in nature. The term 'transient states' would always have
to be accompanied by an explanation of which state is meant. For example, Arctic shrub and
tundra within the boreal-tundra transition zone may be in a transient state, while the dense
boreal forests may be in a transient state in the south.

Line 81: Provide a short justification for why "We expect needleleaf forests in the south to sustain greater
tree cover losses than broadleaf mixed leaf forest."

>> We provided a brief explanation for our expectation (lines 122-123).

Line 139: The authors state "In the absence of fire and harvest, boreal tree cover increased across most
of the boreal distribution range. This increase was lower in the south than the north." Please provide a
percentage rather than saying "most" and provide numerical summaries for the south versus north.

>> We have provided these summaries here and throughout the results section, following the
general comments by reviewer 1 about numerical summaries above (line 207-208).

Line 232: Northern expansion could also be limited by cold, nutrient poor soils (Ellison et al., 2019; Sullivan
et al., 2015) and herbivores (Olnes et al., 2017).

>> We have incorporated these additional explanations in the text (lines 324-325).

Line 248: For this subsection, consider comparing with results from Wulder et al. (2020).

>> We appreciate the reference suggestion. As Wulder *et al.* (2020) discuss biomass, we have
included their results in our discussion on carbon implications (lines 445-448 and 460-461).

Line 258: The authors state, "These observations suggest a lower resilience of southern boreal forests to
wildfire perturbations and the faster proposed by recovery of northern interior forest." These are both
substantial conclusions to draw from the current analysis, though I'm not sure the analysis was designed
to directly test these ideas. If the authors want to make these statements, then I would encourage them
to dig more deeply into the literature around these topics and cite that literature herein.

>> We have deleted this part and replaced it with a discussion on how our results on tree cover
change relate to forest recovery in order to follow suggestions by reviewer 2 (lines 349-350
and 354-359).

Line 273: The authors state, "Tree cover changes are strongly linked to climate." However, the strength
of this relationship is not clearly demonstrated in the results. How much of the variability in tree cover
changes was explained by climate when using the GAMMs? Consider adding this information to the results.

>> The suggested results were included in the supplementary information in a separate file due
to the large number of statistical output from all models.

Line 280: But also see Girardin et al. (2016).

>> We included this important publication in the text and added a brief description (lines 406-
407).

Line 290: Shrubs have certainly expanded in the Arctic, but there is limited evidence for actual northward
shrub expansion (Myers-Smith & Hik, 2018).

>> We thank reviewer 1 for this addition and have altered the text accordingly (line 415-416).

Line 296: ... "in combination with losses related to higher rates of warming..." This pattern isn't evident in
Fig 5 which rather seems to suggest maybe even slightly increasing tree cover in areas that warmed more
rapidly.

>> Reviewer 1 points out an indeed factually incorrect statement. What we meant was that
further warming at the warm margin of the boreal forest could further increase tree cover
losses. We have adjusted this section accordingly (lines 421-424).

Line 321: See also Koven (2013)

>> We have included this helpful reference in lines 461-462.

Line 325: This paragraph about albedo doesn't have a single reference...

>> We have added references to the paragraph (lines 470-476).

Figure 2:

>> We have changed Figure 2 to reflect reviewer comments. It now only depicts absolute tree
cover changes and a map with numerical changes of tree cover across the North American
boreal forests and bigger font size.

- It's important to show the magnitude tree cover change (%) instead of just gain and loss.

- Panel C is missing its panel letter

- Could the perceived decrease in tree cover beyond tree line be due to noise in the tree cover data set?

The MODIS satellites have experienced sensor degradation and orbital changes over their few decades of

operation that influence measurements of surface reflectance, spectral indices, and almost certainly

derived products like tree cover. If tree cover is very low (e.g., say 2%), then it would only take a very

small amount of error in tree cover predictions to cause the perceived decline (e.g., say 1%).

>> This is indeed a very important consideration. In the new map it becomes clear that those

changes are in fact low in magnitude and are usually non-significant. Those trend estimations

likely stem from observation variability of the sensors. The more important conclusion from this

is the lack of tree cover gains beyond the northern boundary.

- The legend key is obstructing the x-axis

- The text size is very small throughout the figure. Consider making all bigger.

- Do the black lines in panels a and b convey the average change that was then smoothed using a running

mean?

>> That is correct.

- The standard error bands should be symmetrical around the black line, correct? However, across much

of the figure the error band appears obstructed by the solid polygon.

>> This is indeed a correct statement. The errors bands are visible at least on one side of the

line representing running means. As they are symmetrical, the corresponding opposite side is

implied. For aesthetical reasons, we have omitted error bands where they overlap with the

coloured areas.

Figure 4. Spell out SBD and TC in the axes.

>> Changed accordingly.

Ellison, S. B., Sullivan, P. F., Cahoon, S. M., & Hewitt, R. E. (2019). Poor nutrition as a potential cause of

divergent tree growth near the Arctic treeline in northern Alaska. *Ecology*, 100(12), e02878.

Girardin, M. P., Bouriaud, O., Hogg, E. H., Kurz, W., Zimmermann, N. E., Metsaranta, J. M., . . . Büntgen,

U. (2016). No growth stimulation of Canada's boreal forest under half-century of combined warming and

CO₂ fertilization. *Proceedings of the National Academy of Sciences*, 113(52), E8406-E8414.

Harsch, M. A., Hulme, P. E., McGlone, M. S., & Duncan, R. P. (2009). Are treelines advancing? A global

meta-analysis of treeline response to climate warming. *Ecology letters*, 12(10), 1040-1049.

Koven, C. D. (2013). Boreal carbon loss due to poleward shift in low-carbon ecosystems. *Nature*

*Geoscience*, 6, 452-456. doi:10.1038/ngeo1801

Mekonnen, Z. A., Riley, W. J., Berner, L. T., Bouskill, N. J., Torn, M. S., Iwahana, G., . . . Grant, R. F.

(2021). Arctic tundra shrubification: a review of mechanisms and impacts on ecosystem carbon balance.

*Environmental Research Letters*, 16(5), 053001.

Myers-Smith, I. H., Forbes, B. C., Wilmsking, M., Hallinger, M., Lantz, T., Blok, D., . . . Lévesque, E. (2011).

Shrub expansion in tundra ecosystems: dynamics, impacts and research priorities. *Environmental Research*

*Letters*, 6(4), 045509.

Myers-Smith, I. H., & Hik, D. S. (2018). Climate warming as a driver of tundra shrubline advance. *Journal*
*of Ecology*, 106(2), 547-560.

Olnes, J., Kielland, K., Juday, G. P., Mann, D. H., Genet, H., & Ruess, R. W. (2017). Can snowshoe hares
control treeline expansions? *Ecology*.

Pan, Y., Birdsey, R. A., Fang, J., Houghton, R., Kauppi, P. E., Kurz, W. A., . . . Canadell, J. G. (2011). A
large and persistent carbon sink in the world's forests. *Science*, 333(6045), 988-993.
doi:10.1126/science.1201609

Rees, W. G., Hofgaard, A., Boudreau, S., Cairns, D. M., Harper, K., Mamet, S., . . . Tutubalina, O. (2020).
Is subarctic forest advance able to keep pace with climate change? *Global Change Biology*, 26, 3965-3977.
doi:10.1111/gcb.15113

Sexton, J. O., Noojipady, P., Song, X.-P., Feng, M., Song, D.-X., Kim, D.-H., . . . Pimm, S. L. (2016).
Conservation policy and the measurement of forests. *Nature Climate Change*, 6(2), 192-196.

Sullivan, P. F., Ellison, S. B. Z., McNown, R. W., Brownlee, A. H., & Sveinbjörnsson, B. (2015). Evidence
of soil nutrient availability as the proximate constraint on growth of treeline trees in northwest Alaska.
*Ecology*, 96(3), 716-727. doi:10.1890/14-0626.1

Wulder, M. A., Hermosilla, T., White, J. C., & Coops, N. C. (2020). Biomass status and dynamics over
Canada's forests: Disentangling disturbed area from associated aboveground biomass consequences.
*Environmental Research Letters*, 15(9), 094093.

Reviewer #2 (Remarks to the Author):

The authors utilize time series of tree cover and disturbance data to characterize changes in the distribution
of boreal forests in Canada and identify two key findings: the southern boundary is experiencing faster
tree cover decline than the northern boundary's tree cover is expanding, resulting in a net reduction in
boreal forest coverage, and that the rates of tree cover changes are related to climate change, wildfires,
and timber harvest. This is an interesting and significant question, with implications, as the authors point
out, for the global carbon-climate system.

>> We appreciate that reviewer 2 acknowledges the importance and implications of our study.

I am broadly supportive of the line of inquiry pursued by the study. This is a key question with lots of
circumstantial or smaller scale evidence, and a continental-scale examination of boreal biogeography is
becoming more feasible with recent advances in remote sensing and computational power. However, the
study is marred by several crucial flaws related to its framing, methodology, and reporting of results, and
I am not sure it is either accessible to a general audience or detailed enough to be credible.

>> We thank reviewer 2 for the support and for recognising the importance of the research
questions we addressed. We also appreciate the very detailed comments and feedback in
regards to a lack of contextual, methodological and reporting detail. We hope we have
addressed these comments accordingly to strengthen our manuscript.

Overall , I found that the paper was clearly written, but also found it lacking major holes in its discussion
of the ecological processes and the relevant literature. I was surprised that the introduction had little
description of the relatively deep literature regarding Arctic-boreal greening trends in North America. Many
others have described how these positive trends in ecosystem productivity have been identified using time
series of a unitless index, the normalized difference vegetation index, but one of the challenges here is
translating the trends in an index to actual ecological change. To me one of the key potential advances of
this paper is the examination of such trends using an ecologically/physically meaningful quantity, tree
cover, which is closely related to (but not exactly the same as) relevant values like carbon storage, carbon
flux, albedo, and more. I was disappointed that the authors seemed to gloss over this, beyond failing to
provide to a general readership the basic framework for our understanding of boreal forests and global
change. Similarly, several key ecological processes, such as the importance of changing boreal forest types
or the likely large importance of post-disturbance recovery, are referenced in the results and somewhat in
the discussion, but the general readership may be lost as to the relevance of these factors considering
their lack of mention in the introduction. The research questions are great, but little is provided to place
them in context for the reader beyond the somewhat overstated boreal forest changes. To summarize, I
recommend including at least one paragraph in the introduction and elaborating on the discussion to
provide a more complete framework of boreal forests and global change.

>> We are very grateful for the supportive comments by reviewer 2 in regards to clarity in
writing, research questions and methodology. We understand from the further comments that
reviewer 2 suggests additional explanations concerning the following aspects of our work:

(1) Reviewer 2 agrees with us about the importance of using quantitative metrics of forest
change that are ecologically interpretable. As stated by the reviewer, we have
highlighted the advantage of our approach compared to studies on vegetation greenness
in the discussion. We do, however, see the benefit of providing context of such an
advantage in the introduction and a more extensive description in the discussion. We
have included such statements accordingly (lines 95-104 and 330-335).

(2) Reviewer 2 recommends to expand our introduction to include further details on
ecological processes underlying boreal forest change. We aimed for a brief and concise
introduction of our research topic but agree that the introduction can be more detailed.
We now cover the role of tree cover in forest dynamics (lines 108-111), processes of
disturbance recovery (lines 50-63) and shifts in forest composition and type (lines 53-
60). We have included these contexts accordingly and have also added to the discussion
of these topics (lines 349-366 and 428-438).

A somewhat related point - I found that the way the authors wrote the paper was a bit sensational. For
one, it rubbed me the wrong way that the abstract and title are referencing "North American boreal forests",
but in reality the study focuses solely on Canada. Excluding Alaska is not necessarily a problem, but it
does represent quite a large part of the boreal North America, and the paper implies it is representing
boreal forests across North America. I suggest toning down scope of the title and abstract so readers are
not misled.

>> Reviewer 2 points to the same issue as reviewer 1 in regards to the lack of Alaskan data.
Alaska certainly forms an important part of North American boreal forests. Changes in Canadian
boreal forests, making up the vast majority of boreal forests in North America, can be
considered to have strong implications for the boreal biome on a continental scale. However,
we have now included tree cover dynamics for Alaska as described in response to comments by
reviewer 1 above. We have also toned down the way we report and interpret our results.

Second, I had several issues with the methods used. We are living in an age where continental scale
remote sensing with medium spatial resolution data over a relatively short time series is widely accessible
through platforms like NASA's data portals and computational tools like Google Earth Engine. Why limit
the analysis to the average over 10,000 plots covering 36 km² each? The boreal forest experiences very
spatially heterogeneous, sporadic, and localized disturbances that are hard to characterize so simply with
such large areas and spanning just 6% of the domain. This might be okay if the only process in question
was something more broadly distributed, such as climate effects. But the large importance of disturbances,
which are relatively sporadic and localized in nature, means that a relatively sparse sampling (covering,
as the authors state, just 6% of the domain), could miss lots of the disturbance-driven tree cover losses
and I am not convinced that the distribution is representative of the disturbance regimes. Especially
considering the missing harvest data for 2016-2019. Because the disturbances play a key role in the boreal
forest and in the study's results, I found the resulting analysis hard to believe. This is especially true given
the exceedingly small rates of tree cover change (roughly 0.5% per year... or 10% over the study period,
feels rather smaller magnitude than I'd expect), despite the occurrence of stand-replacing disturbances.
The approach needlessly discards lots of information, when a fundamentally biogeographic question should
be more geographically comprehensive and include some sort of area estimation. A more complete analysis
would also bolster the climate modeling section - I found the GMM results a bit hard to follow since the
relationships between tree cover change and climate looked rather unstable and perhaps overfit.

>> We appreciate the discussion of our method by reviewer 2 but disagree with the conclusion.
The use of all data points, as suggested, will highly increase the risk of pseudo-replication in a
spatially explicit context. Spatial autocorrelation will likely be a relevant issue to be dealt with.
While there are now statistical methods to incorporate such statistical problems (and we have
used one of these to deal with autocorrelation), we have decided for a sampling approach which
is often used in such studies²⁻⁴. Our nearly 13,000 sample plots cover the entire length of each
south-north transect, transects are located randomly across the entire biome and cover the
major boreal ecoregions as well as areas with different levels of disturbance intensities. As we
use a large and random selected set of samples, we are not certain why reviewer 2 believes
that our approach is not representative of the entire biome. We are happy to respond to more
specific arguments in this respect, if necessary.

As for the missing harvest data, we were able to receive an updated dataset which now spans
the entire study period.

I was sad to see a lack of a map of the actual trends in tree cover change. It is an interesting data product
that I believed the authors have generated, but it is hard to evaluate when it is never shown. Instead, we
are able to see largely aggregated/averaged values over certain subdomains that discard much of the
interesting geographic variability. Showing a binary 'gain/loss' map, especially without regards to
statistical significance or magnitude, is a regrettable simplification of the changes in the boreal forest.

>> We appreciate the enthusiasm of reviewer 2 for comprehensive maps which we certainly
share. Because of our approach to use sample plots along south-north transects (see further
details on our justification above), our results on tree cover change did not cover the entire
spatial extent of the North American boreal forests. To provide spatial context to our results,
we initially decided to include an admittedly coarse map of directional change which broadly
resembles the patterns in tree cover change from our sample plots. After careful consideration
of the valuable comment of reviewer 2, we have now included the suggested map, albeit at a
slightly reduced spatial resolution (Figure 2). While this map provides limited insights in
addressing our research questions, we believe that it forms an important product for further
research and for readers interested in spatially explicit results on tree cover dynamics.

And finally, somewhat reiterating a point about the introduction - what about disturbances prior to 2000?
There are many in this domain and recovery from these disturbances must be structuring a very large part
of the tree cover growth in the boreal interior. I was surprised to see it described simply as "undisturbed"
and still that it received relatively little coverage in both the introduction and the discussion. Some mention
is made of forest resilience to these disturbances, but to me it is a much larger and more significant story
than the relationship with climate. There is an interesting avenue to pursue by using the disturbance data,
which goes back to 1985 rather than 2000, to characterize post-disturbance recovery more specifically.

>> We thank reviewer 2 for this important comment which resembles that of reviewer 1. Please
see our comments above regarding this issue. We have followed both reviewers' advice: We
now distinguish and discuss disturbance impacts prior to 2000.

And relatedly - why use a static land cover map? A key part of the analysis is stratifying and filtering by
land cover type, but a map for just the year 2015 was used. If disturbance and recovery are key parts of
the interpretation of the results, I would expect land cover change to also be significant. How can you
incorporate land cover change, which almost certainly occurs in boreal wildfires and timber harvest, into
the story?

>> We highly value the suggestion by reviewer 2 to use a dynamic land cover map in our
analyses instead of a static map, referring to dynamic land cover changes that can help the
interpretation of tree cover changes. Static maps have commonly been used to attribute boreal
forest characteristics or change to land cover types and vegetation functional groups^{3,5,6}. At the
same time, studies emphasising dynamic changes in land cover have found changes to occur
over considerable areas within the boreal biome⁷. We have given the question of dynamic versus
static land cover considerable attention and believe that it is mainly a question of definition and
purpose. Dynamic land cover changes, such as those addressed by Wang *et al* (2020), consider
a shift from evergreen forests to deciduous forests following fire as land cover change, even
though the deciduous stage may be successional and may eventually be replaced by evergreen
forests⁷. As reviewer 2 points out, considering dynamic changes in land cover can provide

helpful context for observable changes in tree cover or other forest characteristics. However,
such temporary changes require considerable efforts to be detected and can be falsely
interpreted as permanent land cover changes. Static land cover maps consider such changes
only to be relevant when there is a fundamental and permanent change in land cover, for
example deforestation by agricultural expansion or urbanisation. These changes are relatively
uncommon in the boreal biome. For the purpose of our study, we tested the idea that deciduous
or mixed forests benefit from past climate change compared to evergreen forests and may
eventually shift boreal forest composition. We therefore used a static land cover map to
distinguish between tree cover changes of different land cover classes, mainly deciduous versus
evergreen, whereby we deemed permanent land cover more important than dynamic changes.
We have made this purpose and justification clearer in our manuscript (lines 587-592). We
agree with reviewer 2 that the consideration of dynamic changes in land cover is a very
intriguing topic for further research.

In summary, I believe the authors are pursuing an interesting research question that will be of interest to
a potentially large audience of ecologists, global change biologists, and Earth system scientists, but that
the study as presented requires a rework of its framing, a more complete method, and a better description
of its results. In its current state it feels somewhat incomplete, and not an especially robust addition to a
research question that has a lot of activity. It is not ready to be published in Nature Communications as it
is now.

>> We thank reviewer 2 for kind and supporting words regarding our research questions and
implications. We hope that we have addressed all comments and feedback to strengthen our
manuscript.

Some more line and figure specific comments follow:

Line 22: more specific to say "latitudinal" rather than "geographical"

>> The term 'latitudinal' would be misleading here, as we have emphasised locations relative
to the southern and northern boundaries of the boreal biome and not in terms of latitudes. We
explained the reasoning in our methods. To be more specific, we have now used the term 'north-
south' (line 23).

Also line 22: I would put in parentheses the years covered after two decades (2000-2019)

>> Changed accordingly (line 22).

Line 24: A bit confusing how this is written. I think it's clearer to say "Tree cover losses due to wildfire and
timber logging accentuated the tree cover losses in undisturbed region..."

>> Changed to 'substantially reduced' (lines 24-25).

Line 25: is resilience really a part of this paper?

>> We have changed this to 'recovery', as resilience can indeed be interpreted in a broader
context (line 26).

Line 27: Can you be more specific about the impacts of a boreal contraction? It's not that compelling to
say "with implications for...".

>> The word limitation for the abstract of 150 words restricts the extent to which the
implications of our study can be made explicit. We have now rephrased this part to be as specific
as possible (lines 29-30).

The abstract is a bit vague and a bit weak. Including some of the key quantities that you calculated would
be helpful.

>> We now included some quantities and altered the abstract to strengthen it (lines 24-25).

Lines 32-36: The first few sentences of the introduction feel out of order. I would say that boreal climate
change is twice the global average before saying that this climate change is threatening the integrity of
the boreal biome.

>> We have changed the order accordingly (lines 36-37).

Lines 32-41: This is a general audience , and as such I feel it is important to summarize briefly what are
some of the implications for carbon and climate for changes in boreal forest extent. Why should the average
reader care? For example, forest loss and disturbances are reducing the boreal carbon sink in North
America (see the recent paper in Nature Climate Change, Wang et al 2021). It should be brief, but clear...
there are plenty of angles for this context.

>> This is a very important comment and specifies to some extent the general comments of
reviewer 2 above which suggest a more complete description of boreal biome processes and
implications. We have added a concise section on potential consequences for a boreal biome
contraction in the introduction, albeit at a later position than suggested by the reviewer (lines
84-94).

Lines 42-51: This paragraph is well-written. Good job! One comment - I would like to see a citation for the
statement that the conditions in the south will become too hot and dry for boreal tree species. Would that
lead to mortality? Or slower recruitment/establishment? It's a little bit vague, in contrast to the northward
expansion part.

>> We thank reviewer 2 for their supportive words about this section. The citation is already
included in the text. D'Orangeville *et al* (2018)⁸ have demonstrated that an expected warming
of the eastern North American boreal forests leads to reductions in tree growth rates of multiple
species along the southern distribution range. These reductions are accelerated when
precipitation declines. We have now included a more recent experimental study by Reich *et al*
(2022) reiterating previous results and specified the expected consequences of climate change
(lines 54-59).

Line 52: remove the word 'differently'

>> Edited accordingly (line 64).

Line 70: For a general audience, define MODIS . You might also mention here that it is time series data.

>> We now clarified the MODIS data for a general audience (lines 105-106).

Line 71: Your study is really more about Canada than North America broadly. At this point I was expecting
something broader and was confused later when only Canada was studied. Make it clearer here.

>> This comment relates to general comments by reviewers 1 and 2 on the focus of our study
on Canada. We have addressed these comments above, kept the term 'North America' but
specified that we were focussing on Canada and Alaska (line 107).

Lines 78-82: The hypothesis that land cover type is an important factor is very interesting, but was not
set up. This will be confusing for non-specialists. Consider a brief discussion of why these forest types are
important earlier in the introduction.

>> We have included a description of these compositional shifts both in the introduction and as
a brief reminder in the hypotheses (lines 59-63 and 122-123).

Figure 1: The conceptual diagram looks nice, but I am not sure that it is necessary. It's quite busy, so I
found it confusing at first. The core concepts were already well-explained in the text. It is also a bit
unintuitive that the northward expansion is described in a left-right manner. But overall I would remove
it.

>> We appreciate that reviewer 2 finds our conceptual diagram aesthetically pleasing and the
underlying introduction clear. We believe that this diagram acts as an alternative to the
introduction in the text, especially for those readers familiar with the specific processes
described in the main text. Since we are far from reaching the maximum number of figures, we
decided to keep it in the introduction, as we feel many readers will appreciate it as a concise
overview of the dynamics we attempt to study.

Line 104: mean tree cover? I assume you mean for 2000? This is not described in the methods. Were tree
cover changes calculated as just the difference between 2019 and 2000, or as a trend fit to time series?

>> We have clarified the calculation of mean tree cover in the text: the mean tree cover of all
582 years between 2000 and 2019 (lines 144-145). The use of only one year to calculate relative
tree cover change (e.g. the year 2000) is problematic, as the MODIS data is quite variable
between individual years. To account for this variability we decided to take the mean of all years
within our study period.

The calculation of tree cover change is, however, described in detail in the methods 'Tree cover
estimates and change' (lines 549-564).

Figure 2: Why not show the trends in tree cover change?? It is challenging to interpret the map because
of the binary nature of it. Are there really no regions that are neutral or near neutral? For example, it is
confusing to see losses recorded in the Mackenzie mountain range, which has very limited tree cover. It
would make the importance of fires and other disturbances much clearer to see their relative impact on
some of these values. Also - why not label the map as panel C? It was hard to find its description in the
caption. It would also help to see the line indicating the "0" standardized boundary distance (e.g. exactly
halfway between the boundaries) across the map.

Showing both panels A and B seemed unnecessary, especially because you state in the text that because
the results are relatively similar you stick to absolute tree cover changes.

Last note about Figure 2: I recommend using a different projection for displaying areas in this part of the
world. Using a Mercator projection greatly distorts areas at high latitudes, making this map difficult to
interpret when we are comparing areas.

>> We highly appreciate suggestions about our key figure (Figure 2). We have remove panel B
and focussed on absolute tree cover change. We have labelled the map as 'B' for better
readability and have adopted the excellent idea to include the 0-line of the standardised
boundary distance. We have also changed the projection to North America Lambert Conformal
Conic.

All other comments somewhat connect to the general comments of reviewer 2 regarding a more
specific map of tree cover changes. We have addressed these suggestions under the respective
comments above and included such a map.

Lines 135-137: I do not think that you can claim much of the boreal forest is truly 'undisturbed'. You should
qualify that these areas have not been disturbed since 2000 (as you state in Figure 3)

>> We have specified that these areas are undisturbed since 1985, as we have also followed
the advice to distinguish between disturbances 1985-2000 and post-2000 (lines 200-204, see
further comments above).

Figure 3: This was an informative figure. In Panel B, I found it very curious that tree cover should increase
so much in the northern part of the domain for both burned and harvested forests. Why aren't any of the
fire trends negative here, yet many negative values occur in the southern parts of the domain in Panel A?

>> We are happy that reviewer 2 finds our figure informative. We agree that our results for the
northern boreal biome following disturbance are interesting. Our thoughts about why these tree
cover increases occur are found in the discussion but we are open to alternative suggestions.
The second question is unclear as to what it refers to. We assume it is reiterating the point of
the first question, comparing the results of the northern and southern tree cover changes under
the influence of disturbance. We believe the underlying assumption is that tree cover changes
across the biome should be negative following disturbance. As this is comparable to comments
by reviewer 1, we refer to our response further above (lines 69-106). If we mis-interpret the
comments of reviewer 2 here, we kindly ask for a clarification.

Figure 4: Cool figure. I would recommend spelling out standardized boundary distance on the Y axis, since
you have space.

>> We thank reviewer 2 for the compliments and have spelled out the axis title accordingly.

Lines 188-189: Why analyze climate trends across 1980-2019? Isn't this extending beyond the limits of
the tree cover data?

>> We have included two periods of climate trends here, one of which indeed extends back to
prior 2000. We explain the reasoning behind this in the methods: 'Both periods are potentially
relevant for changes of tree cover across the boreal biome. The long-term trend covers a period
of time in which climatic changes became increasingly more observable in high latitudes and
may have therefore influenced biome processes resulting in potential long-term effects on tree
cover changes. The more recent time period is likely to have directly impacted tree cover
changes within our study period.'

Line 229: Why should this contraction of forests be transient? It's not clear to me. If disturbances occurred
frequently enough to cause some of this southern decline in 2000-2019, why shouldn't we expect
disturbances to continue to reduce tree cover going forward? We have already seen major fires in Canada
since 2019 - if anything, I might expect the contraction to accelerate.

>> We believe that reviewer 2 makes an excellent point here regarding the prospect of an
intensifying contraction in the future. However, a resulting accelerated future contraction
would mean that the currently observed contraction is still a transient trajectory. The transient
nature of the contraction would only be resolved, if the northward expansion into the Arctic
matches the speed of biome loss in the south (i.e. reaching equilibrium). We present limited
evidence that this may not (yet) be the case, making transient dynamics more relevant than
the theoretical equilibrium which is still dominant in many models. The resulting contraction
remains transient irrespective of whether an equilibrium may be reached or disturbances lead
to an accelerated contraction. The likely increasing negative impact of disturbances in the

future, as pointed out by reviewer 2, could turn the transient contraction into a long transient
which would indeed accelerate the mismatch between northern expansion and southern retreat.
We attempted to clarify the importance of transients, as described above, in the introduction
and discussion but would be grateful for suggestions where in the text we could be clearer.

Lines 237-246: Thank you for bringing up, albeit briefly, the literature on Arctic/boreal greening trends. I
think this could be discussed further, both in the introduction (to give context) and here, to explain more
why your results are novel. I see the novelty in using physical units to describe greening trends.

>> We are very grateful for reviewer 2 recognising the novel aspect of our study. As pointed
out in the general comments above, we have expanded our introduction and discussion of
various aspects of boreal forest change, including the difference between greening/browning
and tree cover change.

Line 252: I don't know that the citation listed here, Sulla-Menashe et al, really supports this discussion of
disturbances driving forest loss. Sulla-Menashe et al were more describing trends in greenness or
productivity here, which is not necessarily the same thing as changes in forest cover or biomass. I would
find something more directly relevant.

>> We have added other citations directly relevant to forest loss but kept the initial citation, as
greenness trends have been linked to forest loss but are indeed much more challenging to
interpret on their own (line 349).

Lines 256-258: To evaluate these comparisons in wildfire regimes, it would be necessary to provide some
sense of the distribution of fires that occurred in your domain going back further than 2000. I believe that
a number of major fires occurred in the 1990s, for example, in the interior of western Canada, while the
period from 2000-2013 had relatively lower rates of fire. The increased interior tree cover may be more a
matter of sampling than a real signal of different forest resilience to disturbance. Overall, there seems to
be little discussion of post-fire recovery, but I think it is in fact dominating much of the signal in this study.
I think there could be an interesting analysis using the CANLaD data to attribute some of these trends to
post-fire recovery.

>> These valuable comments relate to the point made earlier about including disturbances prior
2000 in the analyses. We have done so and have included a discussion of these results,
especially in the context of post-disturbance recovery (lines 354-370).

Lines 304-305: This statement feels rather speculative, especially without any citations to back up this
notion or elaboration on why this type conversion could happen. I would remove it or expand it, but not
leave it as it is.

>> We deleted the statement (lines 437-438).

Lines 312-324: Much is said here about the losses in tree cover potentially resulting in carbon source. But
the total tree cover increased in the domain, according to your data, which makes me confused why we
are discussing the possibility of a carbon source. Even if it is just small areas that become a source, it
seems to me the bigger story is the total increase in tree cover. This doesn't flow from your results, or I
am missing something.

Also, isn't the majority of the carbon in the boreal biome located in the thick permafrost soils? If we are
discussing carbon budgets in Canada, I do not think this can be ignored.

Line 321-322: There is lots of research investigating disturbance related carbon losses vs warming induced
carbon gains in Canada. I feel that this section is missing a lot of discussion from the literature. It should
be expanded to provide a more nuanced view of the problem.

>> As tree cover can broadly represent biomass and thus above-ground carbon, changes to the
carbon storage ability as a result of changes in tree cover are necessary to discuss. We have
followed the suggestions and expanded the discussion to reflect on potential changes on carbon
at different location within the boreal biome. We have also included a link with permafrost
carbon, albeit brief, as this is not the focus of our research (lines 462-464).

Lines 325-331: Plenty of research discussing changes in albedo in boreal forests that is missing here.

>> We have now added relevant references to this section (lines 470-476) but point out that
large-scale assessments of albedo changes across the boreal biome, which would be relevant
here, are still rare.

Lines 334-338: Regarding MODIS limitations, what about new satellite datasets? Such as new lidar or
higher resolution imagery?

>> New remote-sensing datasets are by nature highly limited in their temporal coverage.
Promising sensors on board Sentinel-2 or the newest generation of Landsat satellites cover less
than 10 years (Sentinel since 2015, Landsat 8 since 2013 and Landsat 8 only since 2021).
Longer-term data sets with higher resolutions, such as Landsat-based time series imagery, are
subject to large uncertainties due to the combination of sensors with different temporal and
spatial resolutions as well as sensor accuracies. Forest cover products with higher resolutions
have been produced, e.g. by Hansen *et al* (2013)⁹, but only provide data on selected years and
over a shorter time span. Finally, Lidar products are highly restricted in their spatial coverage
and cannot yet be used for the continental-scale analyses we conducted here.

New technologies in this regard are nonetheless highly promising and we have included an
additional statement about this in the manuscript (lines 483-488).

Line 374: I am a little confused as to why the data were analyzed in these sample plots. Why not analyze
the entirety of the dataset? Especially considering you are addressing a fundamentally biogeographic
question that is impacted by sporadic, localized disturbances, I would think geographic completeness is
crucial. Certainly, many other studies have analyzed much larger datasets (e.g. at higher resolution and
geographically complete, over longer timescales), so I am not sure that computational limits should explain
this.

>> We appreciate the emphasis reviewer 2 puts on spatial completeness in the data analysis.
We understand that a spatially complete map of tree cover changes are likely to be of interest
to many readers. We have responded to this further above and provided such a map (Figure 2
in the manuscript). As for the data analysis, we refer to our response to the same point in lines
411-423 of this document.

Line 384: I am surprised that you condense each plot into essentially a single value. That is quite a lot of
information being lost, especially since many disturbances and climatic gradients here occur at smaller
scales than 36 km².

>> This comment relates to the general comment by reviewer 2 regarding the sampling
approach and spatial completeness of our study which we have addressed above.

Line 401: What mean tree cover? In 2000, or over 2000-2019?

>> As explained further above, we have clarified in the text that this is mean tree cover of all
of years 2000-2019.

Line 403: You are not calculating biomass change here. Biomass and cover, while sometimes correlated,
are quite different values.

>> We have changed the wording to reflect that the correlation between biomass and tree cover
are not always strong (lines 559-560). For our study, however, there is a very clear correlation
between the two (Figure 1 below, not included in the manuscript). We use the comparison with
biomass in this section of the manuscript to clarify the difference between absolute and relative
tree cover changes. We do not claim that both can be used interchangeably.

Figure 1 Relationship between MODIS tree cover and above-ground biomass for the years 2001 and 2011.
Biomass data was extracted from the Canadian forest attributes map¹⁰. Solid lines represent model fits of a linear
(red) and an exponential model (blue).

Lines 418-420: I looked at Figure S11, and I'm not sure how you can use this to be confident that not
having harvest data for 20% or so of your study period does not change trends. Especially considering
your sampling approach, which might not be representative of the overall geographic patterns of
disturbance.

>> We have now received the latest update of the CanLaD dataset which includes wildfires and
timber harvests from 1985 to 2020. We have updated and re-analysed the data (lines 569-582).
We have also deleted the section on combining CanLaD with the MODIS Burned Area dataset,
as this combination is not relevant anymore. We also deleted associated discussion and figures
in the supplementary information accordingly.

Lines 422-424: I do not think you should call the remaining plots "undisturbed". They have not been burned
or harvested since the year 2000, but most of the forest in this domain will likely have been disturbed
relatively recently before that. You can investigate this with the CanLaD dataset, which includes
observations to 1985. It would at least help us understand a little more about post-disturbance recovery.

>> This comment relates to the interesting idea of reviewer 2 and 1 to distinguish between
disturbances 1985-2000 and post-2000. We have followed this idea now and have changed the
names of categories accordingly to avoid the term 'undisturbed'. See further responses above.

Lines 435-440: Why would the climate change in the time period before your tree cover data have
relevance to the changes in tree cover observed? I do not follow why the 1980-2019 ERA5 data were used.

>> We refer to our response to the comment in line 634 of this document which relates to the
same point.

Lines 456-468: I liked the approach for developing the SBD.

>> We appreciate the supportive statement about the standardisation approach of boundary
distances.

Reviewer #3 (Remarks to the Author):

Generally, the study provides important contribution to the understanding of the interacting influences of
climate change and disturbances to the envisaged northward shift in the spatial coverage of the boreal
biome. It adds to this understanding, a continental scale evidence on percent tree cover changes along a
known south-north gradient of decreasing tree cover. Further, tree cover changes along this gradient are
assessed to show the influence of timber harvest and fire in different vegetation cover types, thus providing
rich information on some of the main interacting factors contributing to the tree cover dynamics in the
biome.

>> We thank reviewer 3 for the supportive words and for acknowledging the importance of our
study.

Suggested improvements:

1. The first line of the abstract reading "Climate change is expected to shift the boreal biome -northward
through expansion at the southern and contraction at the northern boundary respectively." seems to
suggest '... expansion at the northern (not southern as presented) and contraction at the southern (not
northern as presented) ...' unless the direction of expansion/contraction is specified in the current
sentence. Please check and correct as may be relevant.

>> We appreciate reviewer 3 to pointing out this error and have corrected it (lines 19-20).

2. The two decades duration of the study needs to be explained in terms of its potential and/or limitation
in the context of the aims of the study and the influence of disturbances. e.g. how fast are the regeneration
816 times within the biome? would one expect different tree cover trends and thus different conclusions if the
817 MODIS tree cover data set was available for the past 40yrs or 70yrs or so?

>> We thank reviewer 3 for this very valid point. We already included the limitation of temporal
coverage of the MODIS dataset in the discussion where we recommend long-term monitoring
to capture changes in tree cover beyond a time span of 20 years. Following reviewer 3's
comment, we have now specified this limitation (lines 495-497).

3. The two sentences presented in Lines 141 - 144, namely "Losses of tree cover in disturbed areas were
highest at the southern boundary where wildfires and harvests overlap in extent (Figure S3). Interestingly,
tree cover in northern boreal forests increased, despite the occurrence and greater extent of wildfires in
these parts compared to the southern boreal (Figure S3).", provide information on results presented in
both Figure 3 and Figure S3 but only cite Figure S3. To improve on clarity, either the two sentences can
be rephrased to strongly link them with the preceding sentence that cites Figure 3, or Figure 3 can be cited
again in these two sentences.

>> We have now referred to both figures throughout this section (lines 214 and 218).

4. The word "scale" in the sentences "We found that the scale of this absolute change depended on mean
tree cover across our study period" (Line 399 to 400) and "... while relative values are looking at the scale
of change with respect to initial conditions." (Line 404). Does it refer to the temporal scale? or spatial
scale? for the whole study or scale in the sense of size/magnitude of change for each sample plot? Please
clarify as may be necessary.

>> We have now clarified this by removing the word 'scale' which is not necessary as we refer
to the magnitude in both cases (lines 555 and 561).

5. The text suggests a high confidence that missing timber harvest data for the period 2016-2019 does
not considerably change tree cover trends (Lines 418 - 420). This is further elaborated in Figure S11. This
figure, particularly panels A and B suggest "Tree cover trends in the south were generally higher for the
longer time period than the shorter one" (lines 119 - 120 of the supporting file). However, the longer
period (20 years) is just about four years longer than the shorter period (16 years), which may imply that
the four years are contributing some interesting information in the running means such that higher trends
are observed over the longer period. Was the tree cover data for the period 2016-2019 examined
independently? or relative to that of the period 2000-2015 (where timber harvest data is available) or that
of 2000-2019 (the longer period)? Although results in panels C and D justify further analyses without the
missing timber harvest data, it may be interesting to further explain the tree cover data/trends/changes
for this 2016 - 2019 period, particularly because timber harvesting may directly influence: i.the detected
annual tree cover, and trends in tree cover as evident in Figure 3B and ii. arguably, the successive years'
fire behavior.

>> We thank reviewer 3 for the additional idea to investigate a potential effect of missing
timber harvest data for the period 2016-2019. Fortunately, we have been able to update the
dataset on disturbance to cover the missing period.

6. The acronym "SBD" is explained to refer to "Standardised boundary distance" (Line 468 of the main
manuscript document). However, in some cases "standardized boundary position" is used (e.g. Figures S1,
S2, S11, etc.). This raises a question as to whether "Standardised boundary distance" and "standardized
boundary position" are synonymous or they are meant to refer to different characteristics of the sample
plots, as in some cases both are utilized next to each other (e.g. Figure S11). The use of the two phrases
needs to be clarified and the presentation needs to be consistent both in the main manuscript and the
supplementary material document. It is further suggested that, whenever the acronym "SBD" is utilized to
label figure axes (e.g. the y-axis of Figure 4, panel A of Figure S5, Figure S7, panel C of Figure S11, etc.),
it can be written in full either on respective figure's axis or in its caption.

>> We are grateful that reviewer 2 has detected these mistakes in our supplementary
materials. We have used 'position' in an earlier version of the manuscript. Both terms are
interchangeable. We have now unified the terminology to 'Standardised boundary distance' in
the supplementary materials as was already done for the manuscript. We have now also written
out the abbreviations wherever they occurred.

7. The figure label "S4 (B)" is repeated twice while "S4 (C)" is missing. Please correct.

>> We have corrected this accordingly.

8. Future studies may be suggested to look further into changes in species composition within the land
cover types along the south-north gradient, particularly in areas with notable tree cover changes. The
current findings illustrate that timber harvests occur heavily in the south. This may possibly be explained
by higher tree cover in the south, or availability of preferred timber species, etc. In the presence of fire
and other factors (both natural and human induces), different species within the cover types may exhibit
different levels in their regeneration rates, colonization rates, resilience or sensitivity to disturbances, etc.,
which in turn may influence the disturbance patterns and may provide feedback to the transient tree cover
trends at a local- or biome-level scale. Availability of more detailed information on any aspect of these
dynamics may further inform management plans and policies aiming at minimizing carbon sources.

>> We could not agree more with these excellent suggestions by reviewer 3. Increasing the
understanding of processes affecting growth, mortality and reproduction of boreal forests in
the face of climate change and intensifying disturbances will be crucial to predict changes in

forest composition and ecosystem functioning. Detecting species shifts on a larger spatial scale
will, however, be challenging, as field observations over large areas are infeasible, new remote
sensing technologies coupled with detections of machine learning algorithms may be a
promising way forward and could certainly be an interesting pathway for future research.

- 1. DiMiceli, C. M. *et al.* Annual global automated MODIS vegetation continuous fields (MOD44B) at
250 m spatial resolution for data years beginning day 65, 2000–2014, collection 5 percent tree
cover, version 6. Preprint at (2017).
- 2. Staver, A. C., Archibald, S. & Levin, S. A. The global extent and determinants of savanna and
forest as alternative biome states. *Science (1979)* **334**, 230–232 (2011).
- 3. Berner, L. T. & Goetz, S. J. Satellite observations document trends consistent with a boreal forest
biome shift. *Glob Chang Biol* **1–18** (2022) doi:10.1111/gcb.16121.
- 4. Hirota, M., Holmgren, M., van Nes, E. H. & Scheffer, M. Global Resilience of Tropical Forest and
Savanna to Critical Transitions. *Science (1979)* **334**, 232–235 (2011).
- 5. Scheffer, M., Hirota, M., Holmgren, M., van Nes, E. H. & Chapin, F. S. Thresholds for boreal
biome transitions. *Proceedings of the National Academy of Sciences* **109**, 21384–21389 (2012).
- 6. Sulla-Menashe, D., Woodcock, C. E. & Friedl, M. A. Canadian boreal forest greening and browning
trends: An analysis of biogeographic patterns and the relative roles of disturbance versus climate
drivers. *Environmental Research Letters* **13**, (2018).
- 7. Wang, J. A. *et al.* Extensive land cover change across Arctic–Boreal Northwestern North America
from disturbance and climate forcing. *Glob Chang Biol* **26**, 807–822 (2020).
- 8. D’Orangeville, L. *et al.* Beneficial effects of climate warming on boreal tree growth may be
transitory. *Nat Commun* **9**, 3213 (2018).
- 9. Hansen, M. C. C. *et al.* High-Resolution Global Maps of 21st-Century Forest Cover Change.
*Science (1979)* **342**, 850–853 (2013).
- 10. Beaudoin, A., Bernier, P. Y., Villemaire, P., Guindon, L. & Guo, X. J. Tracking forest attributes
across Canada between 2001 and 2011 using a k nearest neighbors mapping approach applied to
MODIS imagery. *Canadian Journal of Forest Research* **48**, 85–93 (2018).

Review comments, second round

Reviewer #1 (Remarks to the Author):

The author's revised manuscript addresses many of the concerns that were originally raised by reviewers. Among other refinements, these included (1) incorporating additional disturbance datasets to identify disturbances from 1985-1999; (2) analyzing spatial variability using ecoregions; (3) providing numerical summaries of results. The analysis and manuscript have been considerably improved. The analysis convincingly shows overall increases in tree cover in the North American boreal forest during recent decades, especially along the northern interior, whereas tree cover decreased near the southern boundary of the biome. These changes are suggestive of a boreal biome shift. However, the headline finding that, "Tree cover changes reveal contraction of North American boreal forests," is less fully developed. The authors state, "The observed regional changes in tree cover resulted in a shift of the boreal distribution boundary and a shrinkage of the expected boreal biome range (Figure S2)." Their new supplemental simulation analysis does support this idea, but for a headline about forest contraction, it seems there would be more information about the spatial extent to which the distribution boundary changed and the domain contracted.

A few further comments:

- The manuscript would benefit from careful proof reading and editing. For instance, there are quite a few sentences that end with two periods or references inserted after the period.
- The results present many of the numerical summaries as percentages, but I believe most are supposed to be percent per year.
- There apparently is a contradiction between the abstract and discussion. The abstract concludes, "The observed asymmetry in tree cover change indicates a possible transition contraction of the boreal biome, which could lead to a net release of carbon..." Whereas part of the discussion reads, "... our results imply carbon uptake of North American boreal forests through an overall increase in tree cover across the biome." My understanding is the later interpretation is more likely.
- The perceived increase in tree cover from 2000-2019 in many northern plots that were disturbed during that period is still perplexing. Perhaps as the authors noted in their response it is related to disturbances only affecting a small fraction of the total area of those plots, with the overall increase in tree cover of undisturbed areas masking the effect of disturbance. Most readers would probably assume that harvested and/or burned plots were fully logged or burned, so it might be good to clarify that plots were flagged as disturbed if there was any (?) evidence of disturbance.

Reviewer #2 (Remarks to the Author):

Overall, I have mixed feeling about the new submission. My main issues are that the authors are presenting very small trends, and their variability/uncertainty should be quantified and report for us to better interpret the results. I also have some questions about how the authors have interpreted some of the results, and how they have responded to some of the reviewer comments. The evidence does not seem very strong for the reported claims - I don't necessarily doubt the results, but I'm not in agreement with the approach. I am not ready to accept the article as is. Please see below.

The authors have gone to considerable lengths to fill out the extent of their analysis by including data for Alaska and expanded disturbance data to help explain their tree cover changes. I applaud their efforts and believe the paper is stronger for it. However, I am concerned that they neglected fires in Alaska, especially as some of the biggest changes in tree cover occur in the northern part of Alaska, primarily as a result of fire. There are plenty of analogous fire datasets that span Alaska

(e.g. MTBS) that could be used to identify fire-affected areas.

The new map in Figure 2 looks really good to me. I know that it does not provide the basis for the authors' statistical inference, but it does provide a clear picture of what is happening in this domain and lends much credibility for the paper. Especially considering that this is a fundamentally geographical problem and because there is so much variability in the intensity of tree cover change since fires happen in localized patches. The reporting of regional scale averages masks a lot of this variability, but the map really illustrates how important fires are for the observed changes.

My main concern about the representativeness of the domain is about the rate of disturbance. Boreal forests experience episodic, large fires that result in a heterogeneous patchwork of forest ages and successional stages at a variety of scales. The way much of this study is presented (e.g. using regional averages) makes it seem like the tree cover change is a gradual, distributed process, when in reality some of the biggest drivers of change are punctuated, localized, and non-randomly distributed. What I wanted to know is whether the rate of disturbance represented by the 7% sample is similar as the rate of disturbance across the domain. I think this can be easily calculated using the available datasets. The sampling approach is defended with its use by other studies; two of those studies are from 40 years ago, before cloud computing is possible. Berner et al's study was less geographic in nature than this one, reporting changes in greenness sampled across a range of environmental conditions, rather than across geographic parameters. I am just looking for a little more completeness, or a more robust defense of the approach.

I thank the authors for putting in the text the average rates of change in latitudinal tree cover extent. I think that it needs to be a little clearer how variable these estimates are. There is no mention of the uncertainty or variability in these average rates of tree cover change; this is especially concerning considering the large variability evident in Figure 2's map and the very small numbers reported. For example, changes on the order of up to a quarter percent per year suggest a total change in tree cover of 5% over the 20 year timespan. This is a fairly small value, considering the large impact that fires can have, and understanding the uncertainty is key to interpreting these values. Meanwhile, DiMiceli et al. (2021) report an RMSE for this data product ranging from 9 to 22% tree cover, which is considerably higher than the average changes that the authors are reporting. It might be more effective to report ranges of this change.

On Figure 3: It confusing to me, as pointed out by Review 1, that fires and harvest should increase tree cover in the northern part of the domain. I am also surprised to find a non-negligible amount of harvest in the northern part of the domain. Especially in combination with my concerns about the accuracy of the data product used, I am concerned about the reliability of the VCF product for this analysis. The authors claim that it is possible for fires to burn at low severity and therefore not reduce the tree cover much - but then, why should the tree cover increase beyond the original fire? Furthermore, it is not common for fires in this part of the region to be of low severity, as there is often limited suppression efforts and usually boreal fires burn quite severely. For example, I suspect many of the fires in the CanLAD dataset are stand-replacing fires that kill most of the trees in an area - or else, it would be hard to detect via Landsat.

I've gotten a little confused about terms, especially after reading the exchange between the authors and reviewer 1. The "contraction" is a change in latitudinal extent of tree cover, yet the tree cover overall increased. But the evidence for a contraction is presented as changes in tree cover in different regions. Is it a contraction of boreal forest extent, or is it a thinning in the south? Much of this hinges on magnitude. Are areas in the south losing forest entirely, or just losing tree cover temporarily? Why is the latitudinal range important? It seems to be just a single dimension of ecosystem change, but the map in Figure 2 demonstrates quite a lot of variability that goes beyond latitude. In fact, most of the variability in tree cover change seems to be regional.

The discussion about land cover types is still confusing to me. The authors dismissed my concerns about land cover change, but disturbance-driven tree cover and biomass changes go hand in hand with land cover change. And increasing body of literature describes interactions between post fire transitions to broadleaf forests and increasing biomass (see, e.g. Mack et al. 2021, Science). I'm not sure how to understand the discussion at the end of the discussion section on environmental conditions. The authors seem to ignore post-fire succession, and I'm not sure how they can

account for it without a dynamic land cover map. Some of the biggest concerns about boreal forests are apparent biome shifts, either via northern treeline advance or via changes in environmental conditions favoring deciduous forests.

Minor comments:

The text on lines 74-80 confused me. The timescales of change in boreal forests is generally pretty slow (100s of years to fully 'recover'), anyway, and given that climate change is an ongoing and evolving process, I'm not sure that there was much discussion out there of this system, or any in particular, being "permanent". I think this 'transients' framework needs to be introduced a little more carefully, because these distinctions between i.e. long and not-long transients don't seem very clear to me. This is especially confusing to me given that the authors dismiss the the importance of land cover change; even if it is transient, climate change may be driving biome shifts that are an integral part of how we understand ecosystem change.

I still find Figure 1 confusing. It feels like there are too many panels to demonstrate a relatively straightforward concept. I feel that it could be displayed in just one. I think panels A through C are meant to show time? But it took me a while to figure that out. Maybe I'm not smart enough for it.

The authors discuss the potential for reduced carbon storage in the long-run due to slow regrowth after disturbance. I'm not sure how the authors reach this conclusion when it appears that total tree cover is increasing, fires and harvest appear to increase tree cover in parts of the domain, and warming temperatures seem to increase tree cover (Figure 5).

Reviewer #3 (Remarks to the Author):

The manuscript has significantly been improved with the i. use of updated CanLaD disturbance dataset, ii. improved figures including the use of a different projection, use of ecotones and generally improved map aesthetics and iii. presentation of information in the manuscript. Additional details provided in the supplementary material document further supports details presented in the manuscript. The authors are commended for the improvements made to facilitate communicating the intended message on this important topic.

Suggested MINOR improvements:

The paragraph on lines 315 - 336 discusses the influence of wildfires on tree cover changes. It is clear that the northern interior of the biome is experiencing more frequent fires but had more gains on tree cover compared to the southern interior with less frequent fires but more losses. In addition to what is presented to explain the faster recovery rates in the north (Lines 331 - 334), it may also be that the more frequent fires experienced in the north are relatively less intense/less severe (given the available fuel) and thus may probably not be intense enough to obstruct regeneration e.g. by inducing mortality on tree's (or woody species') seedlings.

On line 604 - 605: the sentence is probably meant to say all analyses described "above" not "below" as stated.

REVIEWER COMMENTS

Please note: Line indications in our responses refer to the version of our manuscript that includes track changes.

Reviewer #1 (Remarks to the Author):

The author's revised manuscript addresses many of the concerns that were originally raised by reviewers. Among other refinements, these included (1) incorporating additional disturbance datasets to identify disturbances from 1985-1999; (2) analyzing spatial variability using ecoregions; (3) providing numerical summaries of results. The analysis and manuscript have been considerably improved. The analysis convincingly shows overall increases in tree cover in the North American boreal forest during recent decades, especially along the northern interior, whereas tree cover decreased near the southern boundary of the biome. These changes are suggestive of a boreal biome shift. However, the headline finding that, "Tree cover changes reveal contraction of North American boreal forests," is less fully developed. The authors state, "The observed regional changes in tree cover resulted in a shift of the boreal distribution boundary and a shrinkage of the expected boreal biome range (Figure S2)." Their new supplemental simulation analysis does support this idea, but for a headline about forest contraction, it seems there would be more information about the spatial extent to which the distribution boundary changed and the domain contracted.

>> We thank reviewer 1 for their kind words in recognising the improvements of our manuscript which in turn were only possible through the very detailed comments we received earlier. We are also pleased to hear that reviewer 1 is in agreement with most of our results and interpretations.

We are grateful for the critical feedback about a possible contraction of the boreal biome. We appreciate that reviewer 1 agrees with our results from the simulation in the supplementary material. We have interpreted our results in the context of a biome contraction but agree with reviewer 1 that the dynamics in tree cover changes are not a direct measure of such a contraction. However, the observed mismatch between a lack in northern tree cover expansion and southern tree cover loss represents strong structural indicators for the onset of a long-term contraction of the biome. We have now clarified the distinction between tree cover change as a structural indicator of a biome contraction and the long-term contraction itself in the abstract (lines 21-30), introduction (lines 103-108), discussion (lines 309-315) and conclusion (lines 503-509). We have also rephrased the title of our manuscript into a question to better reflect the change in framing.

To provide further strength to our interpretation that tree cover dynamics are indicative of a potential boreal biome contraction, we have followed the suggestion of reviewer 1 for further analyses and information. We have now included three additional lines of evidence in our supplementary materials (from line 72), all of which point towards the contractive nature of tree cover distributions. We find a clear range shrinkage in tree cover distributions and a loss of treed area around the biome boundaries which are further indications that the biome may be

facing a contraction in the long term. We refer reviewer 1 to our description and discussion of these analyses. We furthermore would like to point towards our discussion with reviewer 2 below about the potential future pathways of a biome contraction based on tree cover changes.

A few further comments:

- The manuscript would benefit from careful proof reading and editing. For instance, there are quite a few sentences that end with two periods or references inserted after the period.

>>We thank reviewer 1 for making us aware of the misplaced punctuations and references and have edited the text accordingly.

- The results present many of the numerical summaries as percentages, but I believe most are supposed to be percent per year.

>>The changes are indeed always meant as percent per year and we have added this wherever necessary in the text.

- There apparently is a contradiction between the abstract and discussion. The abstract concludes, "The observed asymmetry in tree cover change indicates a possible transition contraction of the boreal biome, which could lead to a net release of carbon..." Whereas part of the discussion reads, "... our results imply carbon uptake of North American boreal forests through an overall increase in tree cover across the biome." My understanding is the later interpretation is more likely.

>> We agree with reviewer 1 that, based on biome-wide tree cover change, biomass carbon may have increased. However, the short-term change in biomass carbon based on our results from tree cover dynamics may not represent the long-term change in carbon, especially not under a contraction scenario of the biome. The reason for an overall increase in biomass carbon stems from the interior becoming denser. This increase however is limited by the available space and will eventually level off which would likely lead to a net loss of carbon in a contracting boreal biome. We understand that such long-term changes are highly speculative. We have therefore edited the abstract and the associated discussion to distinguish between short-term and potential long-term biomass carbon changes (lines 30-33). Please also note our response to reviewer 2 about potential carbon changes further below.

- The perceived increase in tree cover from 2000-2019 in many northern plots that were disturbed during that period is still perplexing. Perhaps as the authors noted in their response it is related to disturbances only affecting a small fraction of the total area of those plots, with the overall increase in tree cover of undisturbed areas masking the effect of disturbance. Most readers would probably assume that harvested and/or burned plots were fully logged or

burned, so it might be good to clarify that plots were flagged as disturbed if there was any (?) evidence of disturbance.

>> We thank reviewer 1 for this important comment. We have indeed classified all plots as disturbed, if there was any evidence of disturbance. This means that the interpretation for the northern tree cover gains that reviewer 1 provides is the one we think is most likely. While disturbance leads to a considerable tree cover reduction, the surrounding undisturbed areas within our sample plots may be able to compensate these losses on a landscape scale. Our results suggest that such compensation may indeed have occurred in the northern but not in the southern boreal range. We attribute this difference to varying growing conditions between the south and the north. We have extended the discussion on the northern plots accordingly (lines 357-361) and have also clarified the classification of disturbance plots in the methods (lines 575-576).

Reviewer #2 (Remarks to the Author):

Overall, I have mixed feeling about the new submission. My main issues are that the authors are presenting very small trends, and their variability/uncertainty should be quantified and report for us to better interpret the results. I also have some questions about how the authors have interpreted some of the results, and how they have responded to some of the reviewer comments. The evidence does not seem very strong for the reported claims - I don't necessarily doubt the results, but I'm not in agreement with the approach. I am not ready to accept the article as is. Please see below.

The authors have gone to considerable lengths to fill out the extent of their analysis by including data for Alaska and expanded disturbance data to help explain their tree cover changes. I applaud their efforts and believe the paper is stronger for it. However, I am concerned that they neglected fires in Alaska, especially as some of the biggest changes in tree cover occur in the northern part of Alaska, primarily as a result of fire. There are plenty of analogous fire datasets that span Alaska (e.g. MTBS) that could be used to identify fire-affected areas.

>> We are pleased to hear that our efforts in increasing the spatial and temporal scale of our analysis are well recognised by reviewer 2. We are not sure how reviewer 2 gets to the conclusion that the biggest changes in tree cover occur in northern Alaska. It is true that fires have caused considerable losses in Alaskan tree cover based on our map in Figure 2. However, to what extent they compare to other large losses in Canadian boreal forests is not easily interpretable from that map. The statement that we neglected Alaskan fires does not do our efforts justice. We are aware that a large proportion of observed tree cover losses in Alaska are caused by fire. We did not neglect this impact of fire but rather decided to use only one consistent fire dataset for our analyses, i.e. for the much larger Canadian boreal forests. For the sake of spatial completeness, we have now followed the recommendation by reviewer 2

and re-analysed our data by including the MTBS (Monitoring Trends in Burn Severity) fire data. Please see our updated methods for more details (lines 568-572). Contrary to the concerns by reviewer 2, the inclusion of fire data from Alaska did not change the overall patterns of tree cover change we observed in the Canadian dataset (Figure 3 in our manuscript). Similarly, the models of environmental variables did not considerably change the relationships with tree cover trends (Figures 4 and 5 in our manuscript). We agree with reviewer 2 that our analyses are now much stronger, as we showed that observed patterns in tree cover are persistent across both Canadian and Alaskan boreal forests.

The new map in Figure 2 looks really good to me. I know that it does not provide the basis for the authors' statistical inference, but it does provide a clear picture of what is happening in this domain and lends much credibility for the paper. Especially considering that this is a fundamentally geographical problem and because there is so much variability in the intensity of tree cover change since fires happen in localized patches. The reporting of regional scale averages masks a lot of this variability, but the map really illustrates how important fires are for the observed changes.

>> We thank reviewer 2 for the kind words about our map of tree cover change in Figure 2. We agree that this strongly improves the interpretability of tree cover changes we report in a regional context.

My main concern about the representativeness of the domain is about the rate of disturbance. Boreal forests experience episodic, large fires that result in a heterogeneous patchwork of forest ages and successional stages at a variety of scales. The way much of this study is presented (e.g. using regional averages) makes it seem like the tree cover change is a gradual, distributed process, when in reality some of the biggest drivers of change are punctuated, localized, and non-randomly distributed. What I wanted to know is whether the rate of disturbance represented by the 7% sample is similar as the rate of disturbance across the domain. I think this can be easily calculated using the available datasets. The sampling approach is defended with its use by other studies; two of those studies are from 40 years ago, before cloud computing is possible. Berner et al's study was less geographic in nature than this one, reporting changes in greenness sampled across a range of environmental conditions, rather than across geographic parameters. I am just looking for a little more completeness, or a more robust defense of the approach.

>> We thank reviewer 2 for clarifying the comments on the representativeness of our study design and sample plots. We justified our approach to use samples rather than the entirety of available data using important statistical argumentation and referred to recent studies to highlight that a classical sampling design is still used in many studies. All three studies we referred to¹⁻³ are not older than 15 years. An error in the referencing software caused the publication years of 1979 which are incorrect and certainly led to the confusion. We apologise for this. We do not share the view that the study by Berner *et al* (2022) is less geographic. They related trends in greenness to the warmer (i.e. southern) and colder (i.e. northern)

margins of the boreal biome range using environmental variables. This approach is comparable to ours with the only difference that we added an indeed geographical variable with the standardised boundary distance.

Within our analyses we use the raw data extracted from our sample plots, not regional averages as claimed by reviewer 2. All extracted variables within the plots are means but do not represent regional averages. They rather represent landscape-scale parameters of tree cover change and environmental conditions which meet our needs to explore tree cover trends across the North American boreal biome. Wherever disturbances were evident within our sample plots, we classified these plots according to the largest disturbance type (i.e. fire or harvest). The impact of these indeed localised disturbances were evident in landscape-scale tree cover trends. We did not make any claims that these trends represent gradual changes in tree cover. We presented our results as annual trends in tree cover in the presence and absence of disturbances at different times. Such changes include both the disturbance-related losses and recovery/growth.

As for the representativeness of our sample plots, we have calculated the proportion of wildfires and timber harvest within the entire Canadian boreal biome and our sample plots using the CanLaD dataset (see Figure 1 below). The comparison reveals that the proportions are almost identical (Table 1 below) which provides strong evidence of the representativeness of our sampling design for disturbances.

Figure 1 Occurrences of wildfire and timber harvest across the Canadian boreal biome based on the CanLaD disturbance dataset. The black line represents the biome extent. The yellow grids are sample plots within the biome we used in our study.

Table 1 Area extent and proportion of fires and timber harvests within the Canadian boreal biome and the sample plots used in our study (Figure 1 above). The total area represents the area of the entire Canadian boreal biome and the sample plots.

	Canadian boreal	Sample plots
	Area (ha)	
Fire	58,633,211	1,917,950
Harvest	15,877,704	545,202
Total	529,482,711	17,251,036
	Proportion (%)	
Fire	11.1	11.1
Harvest	3.0	3.2

I thank the authors for putting in the text the average rates of change in latitudinal tree cover extent. I think that it needs to be a little clearer how variable these estimates are. There is no mention of the uncertainty or variability in these average rates of tree cover change; this is especially concerning considering the large variability evident in Figure 2's map and the very small numbers reported. For example, changes on the order of up to a quarter percent per year suggest a total change in tree cover of 5% over the 20 year timespan. This is a fairly small value, considering the large impact that fires can have, and understanding the uncertainty is key to interpreting these values. Meanwhile, DiMiceli et al. (2021) report an RMSE for this data product ranging from 9 to 22% tree cover, which is considerably higher than the average changes that the authors are reporting. It might be more effective to report ranges of this change.

>> We thank reviewer 2 for pointing out the variability in tree cover changes across the boreal biome. We have added the uncertainties of mean tree cover changes in the text but are surprised that the standard errors we clearly visualise in all our figures were deemed as insufficient by the reviewer. The numbers mentioned in the text are examples from the gradient of tree cover trends along the south-north gradient depicted in Figures 2 and 3. In addition to the figures in the main text, we provide raw data points from our sample plots in the supplementary materials. We decided to use the average values along the south-north gradient in the main text, as our questions are about the overall patterns in boreal tree cover change moving poleward. As such, a 0.25% annual change in tree cover (i.e. a 5% change over 20 years) in the northern interior, for example, shows that even though some areas lost tree cover by fire (by up to 1% per year or 20% over the study period), tree cover gains through growth or recovery in other areas compensated for these losses. We account for this variability in the additive models for further interpretation of the results.

A comparison of the RMSE of annual tree cover data with the mean trends of tree cover across the biome, as noted by the reviewer, is misleading. As explained above, the mean tree cover trends represent the main trend of all sample plots with a similar relative distance to the boreal boundaries. Naturally, these values are smaller in magnitude than the trends of individual plots or individual pixels in the VCF tree cover dataset. This makes a comparison of RMSE based on pixel values with aggregated landscape-scale tree cover trends meaningless. Nevertheless,

we strongly agree that the product is not free from inaccuracies, as is the case with all remotely-sensed data. Consequently, the trend calculations have to be discussed carefully. That is the reason why we contributed an entire paragraph on the need to validate our results with other remote sensing datasets and field data (lines 473-482). Reporting ranges, as suggested by reviewer 2, would include extreme values in tree cover trends. These extremes would almost always span large ranges of both positive and negative trends (e.g. from -1.5% to +1.5% per year in the northern interior). This gives off the impression that overall change for the northern interior may be net 0 and not, as we show, dominated by tree cover increases.

On Figure 3: It confusing to me, as pointed out by Review 1, that fires and harvest should increase tree cover in the northern part of the domain. I am also surprised to find a non-negligible amount of harvest in the northern part of the domain. Especially in combination with my concerns about the accuracy of the data product used, I am concerned about the reliability of the VCF product for this analysis. The authors claim that it is possible for fires to burn at low severity and therefore not reduce the tree cover much - but then, why should the tree cover increase beyond the original fire? Furthermore, it is not common for fires in this part of the region to be of low severity, as there is often limited suppression efforts and usually boreal fires burn quite severely. For example, I suspect many of the fires in the CanLAD dataset are stand-replacing fires that kill most of the trees in an area - or else, it would be hard to detect via Landsat.

>> We are happy to provide further interpretation to the increasing tree cover in disturbed sample plots between 2000 and 2019. In the previous revision, we have given multiple explanations for these trends. The severity of fire was hereby only one of the options. We agree with reviewer 2 that this one is also not the most likely one given the disturbance dataset. As we have stated in response to reviewer 1 further above, the much more likely possibility, given our results, is that disturbed areas can cover sample plots anywhere from 1% to 100%. In the northern boreal, that proportion on average lies between 20% and 30% (Figure S5 in the supplementary materials). Undisturbed areas within the plots (i.e. the remaining 70%-80% on average) could compensate for disturbance-related losses. It is thus possible and ecologically reasonable that growth stimulation due to improved growing conditions can lead to an increase in tree cover through infilling of available spaces. These increases have the potential to be more substantial on a plot level than the losses incurred by fire or harvest. The remarkable result is that such processes seem to be more dominant in the northern domain of the boreal than the southern domain, even though the proportion of fire disturbance is lower in the south. In our discussion, we have extended the above interpretation of our results. The statement that 'fires and harvest should increase tree cover in the northern part of the domain' is therefore an incorrect interpretation of our results. The more plausible one is that plots that have been disturbed by fire in the northern boreal have increased in tree cover despite the occurrence of fire through possibly faster recovery and compensation from undisturbed areas. We therefore see no reasons to doubt the reliability of the VCF product in this regard. We hope that we could mediate the confusion and concerns.

In regards to the comment about non-negligible harvests in the northern boreal, we are unfortunately unsure what reviewer 2 meant by this statement. Figure S5 clearly shows that harvest occurrences considerably drop north of the boreal interior. This fact is also highlighted by the decrease in harvest data points in Figure S3 and the stop of moving averages of tree cover trends just north of the interior line in Figure 3. In our discussion, we highlight that harvest dominantly occurs in the southern boreal and interpret our results mainly in the context of the southern boreal for harvest-related tree cover trends.

I've gotten a little confused about terms, especially after reading the exchange between the authors and reviewer 1. The "contraction" is a change in latitudinal extent of tree cover, yet the tree cover overall increased. But the evidence for a contraction is presented as changes in tree cover in different regions. Is it a contraction of boreal forest extent, or is it a thinning in the south? Much of this hinges on magnitude. Are areas in the south losing forest entirely, or just losing tree cover temporarily? Why is the latitudinal range important? It seems to be just a single dimension of ecosystem change, but the map in Figure 2 demonstrates quite a lot of variability that goes beyond latitude. In fact, most of the variability in tree cover change seems to be regional.

>> We hope to be able to clear the confusion. In our introduction, we define a potential biome contraction as a latitudinal shift in tree cover distribution which is characterised by a mismatch between a slow shift in the north and a faster shift in the south. Such a mismatch in tree cover distributions would point towards a biome contraction and an ultimate replacement of boreal forests by other systems especially at their southern margin, if this mismatch in dynamics continues into the future. While our study period is too short to show a complete loss of tree cover in the south, the observed thinning along the southern boundary and lack of change at the northern boundary are already strong indicators for the onset of a contraction. We also do not see any reason as to why thinning in the south would be temporary, as reviewer 2 suggests. Reviewer 2 describes in the previous round of comments that drivers of tree cover loss, such as disturbances, are unlikely to decline in magnitude but may rather accelerate. Latest research likewise highlights fire intensification⁴. We therefore believe that the observed contraction in tree cover distribution extent is a precursor for a complete biome loss and shrinkage in the future. We also demonstrate in our discussion and supplementary material that such a contraction and the observed overall increase in tree cover are not mutually exclusive. Both processes can occur in parallel, i.e. the boreal biome becomes denser, while the latitudinal extent shrinks. We also refer to our response to reviewer 1 above and the additional lines of evidence we now include in the supplementary material. All of them point towards a shift in tree cover distributions and a shrinkage of the tree-covered area along the southern and northern biome boundary (lines 72-154 in the supplementary materials). In addition, to avoid further confusion, we have changed the framing of our manuscript to reflect that shifts in tree cover distributions are not a direct measure of a biome contraction but are rather indicators for the onset of a long-term contraction (see updated abstract (lines 21-30), introduction (lines 103-108), discussion (lines 309-315) and conclusion (lines 503-509)). We have also rephrased the title into a question.

The evidence we present is not simply based on 'changes in tree cover in different regions' but rather very much focussed on the boreal boundaries. Here, the first signs of a shift and contraction are to be expected. Reviewer 2 points out in their comment below that 'Some of the biggest concerns about boreal forests are apparent biome shifts, either via northern treeline advance...'. We fully agree with this statement which is precisely why we use the change in latitudinal distribution of tree cover as indicator of such a shift. That is also why the latitudinal extent is so important. The comments by reviewer 2 seem to suggest that we exclusively focus on this latitudinal extent as a measure of ecosystem change. But in fact, we include analyses on biome-wide tree cover change in relation to a range of environmental conditions. Tree cover change is a proxy for many ecologically relevant processes, such as carbon and biomass fluxes, changes in micro and regional climates, plant growth, forest mortality and renewal. We acknowledge that other ecosystem changes, such as shifts in species distributions, are equally important. For our study, however, we clearly defined the focus on tree cover change. Lastly, we are confused about the distinction between latitudinal and regional variability that reviewer 2 mentioned. Why would changes in tree cover along a latitudinal gradient not also be regional? The patterns in tree cover along the latitudinal gradient we find includes many of the tree cover changes in the interior that we think reviewer 2 means with regional variability. We are also not dismissing the considerable tree cover losses within the boreal interior caused by fire or logging. Neither do we ignore the areas of tree cover increases in the mountainous areas along the southwestern boreal biome where forests may be doing better than their counterparts at lower altitudes. In fact, we relate all of these changes in tree cover to a range of environmental conditions, including climate and altitude. The key result of our manuscript, however, is that despite the occurrence of these more localised patterns of tree cover change, a clear biome-wide pattern emerged along the latitudinal gradient which is an indicator for a shift in the biome extent and for a future biome contraction.

The discussion about land cover types is still confusing to me. The authors dismissed my concerns about land cover change, but disturbance-driven tree cover and biomass changes go hand in hand with land cover change. And increasing body of literature describes interactions between post fire transitions to broadleaf forests and increasing biomass (see, e.g. Mack et al. 2021, Science). I'm not sure how to understand the discussion at the end of the discussion section on environmental conditions. The authors seem to ignore post-fire succession, and I'm not sure how they can account for it without a dynamic land cover map. Some of the biggest concerns about boreal forests are apparent biome shifts, either via northern treeline advance or via changes in environmental conditions favoring deciduous forests.

>> We appreciate the discussion with reviewer 2 about land cover changes. We apologise, if we gave off the impression of dismissing the concerns. As pointed out in our lengthy previous response, we do, in fact, strongly agree with reviewer 2 that data on land cover changes would be incredibly helpful to interpret our results on tree cover trends, especially in areas affected by disturbance. However, we do not understand how these potential land cover changes would

affect the clear patterns we observed in tree cover changes across the boreal biome, as indicated by reviewer 2. We do not dismiss the impact on successional changes in boreal forests after fire but are not certain how this would influence the evidence we present in the context of a biome contraction. Besides, creating a dynamic map of land cover change is challenging and would certainly exceed the scope of our study. The difficulty in such an endeavour is apparent in the lack of such maps on a biome scale. Existing efforts usually focus on a smaller spatial extent to map land cover change⁵. To provide some context on land cover impacts on tree cover trends, we included the Global Land Cover product. We are aware that such a static map does not cover the dynamic nature of land cover changes, in particular following fire disturbance. We are thus limited in the extent to which we can interpret tree cover changes in parts of the boreal biome. We do, however, describe possible successional changes towards broadleaf forests in the introduction and discuss these in the context of our results. We are therefore confused as to why reviewer 2 believes we are ignoring such changes in our interpretation.

We agree that land cover changes towards broadleaf forests or temperate shrublands and grasslands are an exciting area of future research, particularly the question how permanent such transitions will be. However, such questions do not form part of our manuscript. The main objective of our work focuses on a potential shift of the biome based on tree cover distributions. The evidence we show from the northern and southern boreal margin, where shifts are most likely to be expected, clearly draws a picture of such a shift. That some of the tree cover dynamics are caused by a successional change to deciduous species is likely based on previous field evidence and regional remote sensing work. However, how widespread these shifts are within the boreal biome and to what extent they constitute either a shift in ecosystems or temporary succession with later replacement by conifers is still very uncertain⁵. Such processes are also unlikely to play out over our relatively short study period of 20 years.

Minor comments:

The text on lines 74-80 confused me. The timescales of change in boreal forests is generally pretty slow (100s of years to fully 'recover'), anyway, and given that climate change is an ongoing and evolving process, I'm not sure that there was much discussion out there of this system, or any in particular, being "permanent". I think this 'transients' framework needs to be introduced a little more carefully, because these distinctions between i.e. long and not-long transients don't seem very clear to me. This is especially confusing to me given that the authors dismiss the the importance of land cover change; even if it is transient, climate change may be driving biome shifts that are an integral part of how we understand ecosystem change.

>> We agree that timescales for biome shifts are a topic of high uncertainty. However, timescales are not the relevant aspect of transient dynamics of biome shifts we tried to bring across. Most ecological processes are transient processes, as system equilibria are rarely reached. In the case of biome shifts, we stress the consideration of transients because they stand in stark contrast to the mainly equilibrium-based predictions of biome shifts which may

not occur because of the described disequilibrium of process rates at the biome margins. We show evidence of such a mismatch in rates based on tree cover change which is an indication for an ongoing biome shift which, contrary to biome model predictions, is contractive. The future trajectories we sketched in the section that reviewer 2 refers to are important as we only studied tree cover dynamics over a relatively short time period which may not necessarily remain the same in the future. There are different imaginable future trajectories which we illustrate in Figure 2 below. (B) represents the continuing contraction of the biome extent which we believe is already ongoing based on tree cover dynamics. The contraction could remain for a long time period or even accelerate, if no or a small northern expansion will occur. However, the mismatch between slow northern expansion and fast southern retreat may be resolved, if a northern expansion of boreal forests into tundra landscapes will speed up. In that case, we will observe a symmetric shift of the biome northward (C). An alternative end to the biome contraction may also be reached, if the drivers of biome loss decline in magnitude and the biome ‘rebounces’ (D). Such a biome recovery could be coupled with lagged northern expansion (E). Based on our results, trajectories D and E are unlikely, as we do not see any reason why the drivers of tree cover loss (i.e. disturbance pressure and warm temperatures) should reverse anytime soon. To avoid confusion about timescales, we have changed the text about these trajectories to emphasise the trajectories, not time (lines 78-88).

Figure 2 Possible future trajectories of boreal biome shifts. (A) The initial latitudinal extent of the boreal biome. (B) A transient contraction of the biome caused by a mismatch between fast biome losses at the southern boundary and a lack of biome expansion at the northern boundary. (C) Northern expansion accelerates to match the pace of southern biome loss, leading to a symmetrical biome shift. (D) Northern expansion does not occur but the southern biome recovers as the drivers of biome loss decrease in magnitude. (E) Similar to D but with a lagged northern expansion.

I still find Figure 1 confusing. It feels like there are too many panels to demonstrate a relatively straightforward concept. I feel that it could be displayed in just one. I think panels A through C are meant to show time? But it took me a while to figure that out. Maybe I'm not smart enough for it.

>> We have changed Figure 1 to make it more intuitive. We still believe that such an overview figure helps readers better understand the possible consequences of mismatches in northern biome expansion and southern retraction. It also provides the base for why we interpret a mismatch as a biome contraction. As correctly pointed out by reviewer 2 in their first review of our manuscript, Nature Communications attracts readers from a wide range of fields and backgrounds. An overview figure may therefore help those readers who do not perceive the described dynamics as a straightforward concept to better interpret our results.

The authors discuss the potential for reduced carbon storage in the long-run due to slow regrowth after disturbance. I'm not sure how the authors reach this conclusion when it appears that total tree cover is increasing, fires and harvest appear to increase tree cover in parts of the domain, and warming temperatures seem to increase tree cover (Figure 5).

>> The discussion reviewer 2 refers to consists of two parts. In the first, we clearly state that based on our results on tree cover change of the past 20 years, biomass carbon on a biome scale has increased. We also refer to existing literature providing evidence of increased uptake. The second part puts these changes in the context of a biome contraction over longer time periods. Here, we caution that the potential uptake of biomass carbon over 20 years may not be indicative for the net carbon balance of a contracting boreal biome in the long run due to expected increases in fire, non-linear dynamics, increased anthropogenic pressure and the partial release of the much larger carbon pool in permafrost soils. As this distinction did not seem to be clear enough, we have edited this paragraph accordingly (lines 439-463).

Reviewer #3 (Remarks to the Author):

The manuscript has significantly been improved with the i. use of updated CanLaD disturbance dataset, ii. improved figures including the use of a different projection, use of ecotones and generally improved map aesthetics and iii. presentation of information in the manuscript. Additional details provided in the supplementary material document further supports details presented in the manuscript. The authors are commended for the improvements made to facilitate communicating the intended message on this important topic.

>> We thank reviewer 3 for the kind words in recognising the improved version of our manuscript which we attribute to the detail comments by all three reviewers in the previous revision round.

Suggested MINOR improvements:

The paragraph on lines 315 - 336 discusses the influence of wildfires on tree cover changes. It is clear that the northern interior of the biome is experiencing more frequent fires but had more gains on tree cover compared to the southern interior with less frequent fires but more losses. In addition to what is presented to explain the faster recovery rates in the north (Lines 331 - 334), it may also be that the more frequent fires experienced in the north are relatively less intense/less severe (given the available fuel) and thus may probably not be intense enough to obstruct regeneration e.g. by inducing mortality on tree's (or woody species') seedlings.

>> Reviewer 3 makes an interesting point indeed. We agree that fire intensity may contribute to an inhabitation of post-fire recruitment. Relatedly, a number of studies has also highlighted the impact of increased fire frequency on seed sources and thus recruitment potential. We doubt, however, that fires have resulted in lower mortality in northern regions, as most fires in the CanLaD dataset are stand-replacing fires with high levels of mortality in the 30m-pixels. In the context of our study, all explanations discussed here, while possible, remain speculative. We have thus limited our discussion to a few points.

On line 604 – 605: the sentence is probably meant to say all analyses described "above" not "below" as stated.

>> Changed accordingly.

1. Hirota, M., Holmgren, M., van Nes, E. H. & Scheffer, M. Global Resilience of Tropical Forest and Savanna to Critical Transitions. *Science* **334**, 232–235 (2011).
2. Staver, A. C., Archibald, S. & Levin, S. A. The global extent and determinants of savanna and forest as alternative biome states. *Science* **334**, 230–232 (2011).
3. Berner, L. T. & Goetz, S. J. Satellite observations document trends consistent with a boreal forest biome shift. *Glob Chang Biol* 1–18 (2022) doi:10.1111/gcb.16121.
4. Zheng, B. *et al.* Record-high CO₂ emissions from boreal fires in 2021. *Science* **379**, 912–917 (2023).
5. Wang, J. A. *et al.* Extensive land cover change across Arctic–Boreal Northwestern North America from disturbance and climate forcing. *Glob Chang Biol* **26**, 807–822 (2020).

Review comments, third round

Reviewer #2 (Remarks to the Author):

The authors' revisions have improved my ability to intuitively understand their results, especially around the reframing of the paper's findings towards a question rather than an assertion. I support the new title. Their responses have generally resolved my concerns. I appreciate the more cautious approach to interpreting small trends, and am largely happy with the modifications and clarifications in the paper. I still have a few issues, but nothing worth blocking the publication of this article. I am in favor of its acceptance. What follows are a few suggestions that I think could still improve the paper, but I do not view as strictly necessary for its publication.

I really appreciate the inclusion of the MTBS. I know that they did not greatly alter the results; however, it was a notable absence that I think readers would wonder about. I really appreciated the figures you present in the rebuttal concerning the representativeness of the samples with respect to disturbance, and my concerns are resolved. I would consider adding this to the supplementary.

I guess I am still confused about the concept that fires and harvest are still generally raising tree cover. Figure 3A makes a lot of sense to me. Figure 3B, less so. I can understand that old fires/harvests lead to recovery, but I am less sure of the recent fires/harvests. In your rebuttal, you note that your sample areas that are recently disturbed can still accrue tree cover due to growth in the undisturbed portion of the samples, especially since those areas may be released from competition. This is not that convincing an argument for me because the samples are rather large - maybe 25 km² in area, if I'm not mistaken. A lot can happen in that large area and it's not intuitive to me that the 20% of forest that burn in one corner of the sample should improve growth prospects in the other parts of the sample. Because I have a much better understanding of the main message of the paper, I'm not sure that it will be worth redoing the analysis to generate a more refined treatment of disturbance. But I am sure that other readers will see this, and wonder. So I suggest adding to the main text to add context or provide some explanation.

The results Figure 5 are hard for me to interpret. To me, the large bulk of the paper is about the observations, not the GAM results. It could be expanded upon in the discussion? It would make sense for me to remove or take it the Supplementary, too.

Line-by-line comments:

Line 30: Here and throughout the paper. You describe your results as indicating the onset of a biome contraction. There is so much uncertainty that I would like it described as "the potential onset of a biome contraction". There are maybe half a dozen instances where I saw this throughout the paper and had this thought.

Line 30-32: This assertion about the biophysical and carbon implications of a boreal forest contraction aren't wrong, but I don't feel that they are supported directly by your results. They are more implications for a discussion section... I suggest removing this from the abstract.

Figure 1 is much more intuitive to me now. I think it's great. Thanks.

Line 386: I found an errant comma at the end of a sentence here

All line references are apply to the marked-up version of our manuscript.

Response to reviewers

Reviewer #2 (Remarks to the Author):

The authors' revisions have improved my ability to intuitively understand their results, especially around
the reframing of the paper's findings towards a question rather than an assertion. I support the new title.
Their responses have generally resolved my concerns. I appreciate the more cautious approach to
interpreting small trends, and am largely happy with the modifications and clarifications in the paper. I still
have a few issues, but nothing worth blocking the publication of this article. I am in favor of its acceptance.
What follows are a few suggestions that I think could still improve the paper, but I do not view as strictly
necessary for its publication.

>> We highly appreciate that reviewer 2 is in favour of the publication of our article. We would
like to express our sincere appreciation and gratitude for all their detailed comments and
feedback which immensely improved our manuscript. It has been a pleasure to work together
on progressing our manuscript.

I really appreciate the inclusion of the MTBS. I know that they did not greatly alter the results; however,
it was a notable absence that I think readers would wonder about. I really appreciated the figures you
present in the rebuttal concerning the representativeness of the samples with respect to disturbance, and
my concerns are resolved. I would consider adding this to the supplementary.

>> We agree with reviewer 2 that the MTBS certainly complemented the spatial coverage of
our study. It is indeed a good idea to include the figures on the representativeness of our
sampling design in the supplementary information. We have followed this advice. The figure
and table now form part of the supplementary information and are referred to in lines 475-476.

I guess I am still confused about the concept that fires and harvest are still generally raising tree cover.
Figure 3A makes a lot of sense to me. Figure 3B, less so. I can understand that old fires/harvests lead to
recovery, but I am less sure of the recent fires/harvests. In your rebuttal, you note that your sample areas
that are recently disturbed can still accrue tree cover due to growth in the undisturbed portion fo the
samples, especially since those areas may be released from competition. This is not that convincing an
argument for me because the samples are rather large - maybe 25 km² in area, if I'm not mistaken. A lot
can happen in that large and area and it's not intuitive to me that the 20% of forest that burn in one corner
of the sample should improve growth prospects in the other parts of the sample. Because I have a much
better understanding of the main message of the paper, I'm not sure that it will be worth redoing the
analysis to generate a more refined treatment of disturbance. But I am sure that other readers will see
this, and wonder. So I suggest adding to the main text to add context or provide some explanation.

>> We agree that the results on tree cover gains in recently disturbed sample plots are curious
but are still convinced that tree cover gains in undisturbed parts of our sample plot in
combination with a relatively fast recovery of the disturbed portion can compensate for the such
losses. A tree cover increase in 80% of the sample plot in combination with a fast recovery from
fire in burnt areas certainly has the potential to compensate losses incurred by fire. We feel
that we have already contributed a fairly long paragraph on possible explanations. We have
now added a few additional sentences for further context lines 302-308.

The results Figure 5 are hard for me to interpret. To me, the large bulk of the paper is about the
observations, not the GAM results. It could be expanded upon in the discussion? It would make sense for
me to remove or take it the Supplementary, too.

>> We understand that the model outputs in Figure 5 may be more difficult to digest. Moving
Figure 5 to the supplementary would, in our eyes, logically also require a similar approach for
Figure 4. In both cases, the results are related to two of our three main research questions.
Figure 5 provides additional support that the observed tree cover trends not only display
geographical patterns but are also related to climatic gradients. A removal of Figure 5 from the
main text would therefore greatly weaken the manuscript. To address the important point
reviewer 2 made, we left Figure 5 in the main text but only displayed the most important
environmental variables and moved the remaining ones to the supplementary. We believe that
it is now much easier to interpret the outcome, yet provide a full set of model results.

Line-by-line comments:

Line 30: Here and throughout the paper. You describe your results as indicating the onset of a biome
contraction. There is so much uncertainty that I would like it described as "the potential onset of a biome
contraction". There are maybe half a dozen instances where I saw this throughout the paper and had this
thought.

>> We generally agree with the rephrasing suggested by reviewer 2 but would like to stress
the importance of clearly and confidently framing interpretations of scientific results. We
strongly believe that our results are a precursor for a future contraction of the biome and
interpret our results in this context. We fully understand that reviewer 2 and perhaps other
readers may disagree with us or at least may be more careful in agreeing with us. However, we
believe that scientific discourse should be fuelled by such differences in the interpretation of
research results. On this occasion and given the relatively short study time period we used, we
are willing to alter the text according to the suggestion by reviewer 2.

Line 30-32: This assertion about the biophysical and carbon implications of a boreal forest contraction
aren't wrong, but I don't feel that they are supported directly by your results. They are more implications
for a discussion section... I suggest removing this from the abstract.

>> We have shortened this part of the abstract to put less emphasis on carbon implications
(line 32).

Figure 1 is much more intuitive to me now. I think it's great. Thanks.

>> We appreciate this comment. We like it too.

Line 386: I found an errant comma at the end of a sentence here

>> We thank reviewer 2 for their keen eyes and removed the comma.